# Integrative Analysis of Transcriptomics and Proteomics for Screening Genes and Regulatory Networks Associated with Lambda-Cyhalothrin Resistance in the Plant Bug *Lygus pratensis* Linnaeus (Hemiptera: Miridae)

**DOI:** 10.3390/ijms26041745

**Published:** 2025-02-18

**Authors:** Jing Chen, Zhi-Jia Huo, Fei-Long Sun, Li-Qi Zhang, Hai-Bin Han, Jiang Zhu, Yao Tan

**Affiliations:** 1College of Horticulture and Plant Protection, Inner Mongolian Agricultural University, Hohhot 010019, China; chenjing666@emails.imau.edu.cn (J.C.); huozhijianmg@126.com (Z.-J.H.); sunfeilong14747@163.com (F.-L.S.); zlq_1232023@163.com (L.-Q.Z.); hhb.25@163.com (H.-B.H.); 2Research Center for Grassland Entomology, Inner Mongolian Agricultural University, Hohhot 010019, China; 3Agricultural Genomics Institute at Shenzhen, Chinese Academy of Agricultural Sciences, Shenzhen 518120, China; zhujiang@caas.cn; 4Key Laboratory of Grassland Resources, Ministry of Education, Inner Mongolian Agricultural University, Hohhot 010019, China

**Keywords:** proteome, transcriptome, lambda-cyhalothrin, resistance, detoxification, *Lygus pratensis*

## Abstract

The prolonged use of pyrethroid insecticides for controlling the plant bug *Lygus pratensis* has led to upward resistance. This study aims to elucidate the molecular mechanisms and potential regulatory pathways associated with lambda-cyhalothrin resistance in *L. pratensis*. In this study, we constructed a regulatory network by integrating transcriptome RNA-Seq and proteome iTRAQ sequencing analyses of one lambda-cyhalothrin-susceptible strain and two resistant strains, annotating key gene families associated with detoxification, identifying differentially expressed genes and proteins, screening for transcription factors involved in the regulation of detoxification metabolism, and examining the metabolic pathways involved in resistance. A total of 82,919 unigenes were generated following the assembly of transcriptome data. Of these, 24,859 unigenes received functional annotations, while 1064 differential proteins were functionally annotated, and 1499 transcription factors belonging to 64 distinct transcription factor families were identified. Notably, 66 transcription factors associated with the regulation of detoxification metabolism were classified within the zf-C2H2, Homeobox, THAP, MYB, bHLH, HTH, HMG, and bZIP families. Co-analysis revealed that the *CYP6A13* gene was significantly up-regulated at both transcriptional and translational levels. The GO and KEGG enrichment analyses revealed that the co-up-regulated DEGs and DEPs were significantly enriched in pathways related to sphingolipid metabolism, Terpenoid backbone biosynthesis, ABC transporters, RNA transport, and peroxisome function, as well as other signaling pathways involved in detoxification metabolism. Conversely, the co-down-regulated DEGs and DEPs were primarily enriched in pathways associated with Oxidative phosphorylation, Fatty acid biosynthesis, Neuroactive ligand–receptor interactions, and other pathways pertinent to growth and development. The results revealed a series of physiological and biochemical adaptations exhibited by *L. pratensis* during the detoxification metabolism related to lambda-cyhalothrin resistance. This work provided a theoretical basis for further analysis of the molecular regulation mechanism underlying this resistance.

## 1. Introduction

From the late 1990s, with the widespread adoption of Bt cotton in China, the tremendous damage by agricultural pests such as the cotton bollworm *Helicoverpa armigera* and other Lepidoptera insects was effectively controlled. The dramatic change in chemical control patterns led to increasing damage by non-target pests such as Mirid bugs [1,2]. The plant bugs *Apolygus lucorum*, *Adelphocoris suturalis*, and *Lygus pratensis* are the most common species, among them, *A. lucorum* and *A. suturalis* mainly distributed in the cotton field of the Yangtze River basin, while the dominated species *L*. *pratensis* severely harms various economic crops such as cotton and alfalfa in the northwestern region [3,4,5,6]. With the intensive cultivation of alfalfa in the pastoral–agricultural transitional zone of northern China and the increasing crop rotation with different varieties, the occurrence and damage of *L*. *pratensis* have been exacerbated year by year [7]. The plant bug *L. pratensis* Linnaeus (Hemiptera: Miridae) is a significant phytophagous pest characterized by its piercing–sucking mouthparts [8]. This pest has four overlapping generations annually in northwestern China [9]. Mirid bugs pierce and suck young shoots and leaves with both adults and nymphs, which seriously interrupts the normal growth of plants and adversely affects the yield and quality of crops [10,11,12]. Currently, the green integrated control strategies in cotton and alfalfa fields primarily depend on color plate trapping [13,14], along with ecological regulation measures [15,16], including regional integrated pest management measures [17] for effective control. However, chemical control was mainly relied on when they seriously broke out, and the long-term irrational application of traditional insecticides such as lambda-cyhalothrin, imidacloprid, malathion, and thiamethoxam has accelerated the resistance development in *L. pratensis* [18,19,20].

Synthetic pyrethroid insecticides are a class of cyclopropane ester insecticides that contain phenoxy groups [21,22], including bifenthrin, lambda-cyhalothrin, cypermethrin, fenvalerate, and deltamethrin [23,24], which exhibit excellent biological activity and high environmental compatibility [25]. Lambda-cyhalothrin has been widely employed to control various agricultural pests [26](Xu et al., 2023), such as the plant bug *Lygus pratensis* [27], the cabbage worm *Pieris rapae* [28], the citrus psyllid *Diaphorina citri Kuwayama* [29], the peach fruit moth *Carposina sasakii* [30], the striped flea beetle *Phyllotreta striolata* [31], the plant bug *Riptortus pedestris* [32], the cotton aphid *Aphis gossypii* [33], and so on. The toxicity assay conducted by Ma et al. (2015) evaluated seven insecticides for effectiveness against *L. pratensis* adults, revealing that cypermethrin and thiamethoxam-HCFC microcapsules have a good insecticidal effect against Mirid bugs [34]. Additionally, Da et al. (2019) assessed the toxicity of six insecticides on the fourth instar larvae of *L. pratensis* and found that lambda-cyhalothrin was the most effective. However, the frequent application of insecticides has resulted in increasing resistance [35]. Zhen et al. (2018) found that seven Cytochrome P450 genes significantly overexpressed in the resistant AL-R population of *L*. *lucorum* [36]. Tan et al. (2021) reported that the resistance of *L. pratensis* to lambda-cyhalothrin, imidacloprid, and other insecticides increased yearly across seven alfalfa fields in northern China [19].

The transcriptional gene expression levels of organisms in specific times and places were determined based on RNA-Seq technology [37]; iTRAQ proteome sequencing is employed to quantify and analyze protein expression levels [38]. The combined omics analysis can obtain a more comprehensive perspective on biological and physiological changes in an organism, enabling the exploration of biological issues from multiple angles. Insects exhibit metabolic resistance to insecticides mainly due to enhanced activities of detoxification enzymes [39]. The detoxification mechanisms for exogenous compounds in insects predominantly involve Cytochrome P450 monooxygenase, Glutathione S-transferase, Carboxylesterase, and ABC transporter proteins [40,41]. These enzymes play critical roles in the detoxification metabolism of synthetic pyrethroid insecticides across various insect species, including the cotton bollworm *Helicoverpa armigera* [42,43], the mosquito *Anopheles sinensis* [44], and the beet armyworm *Spodoptera exigua* [45]. Currently, the combined omics analysis of resistant and susceptible insect strains serves as the primary technical approach for identifying key genes and proteins involved in responses to insecticide stress. For instance, Xu et al. (2020) utilized RNA sequencing (RNA-seq) to analyze field and resistant populations of *Prodenia litura*, revealing that the up-regulation of several P450, GST, and UGT genes was associated with resistance to lambda-cyhalothrin [46]. Similarly, Wang et al. (2022) performed genomic and transcriptomic analyses of *Rhopalosiphum padi*, confirming that the overexpression of *CYP6DC1* or *CYP380C47* contributed to increased resistance to lambda-cyhalothrin in this specie [47]. Proteomic analysis of lambda-cyhalothrin-resistant *Culex pipiens pallens* strain identified several candidate proteins implicated in insecticide metabolism and resistance, including keratin, Cytochrome P450, Glutathione S-transferase, and ribosomal proteins (Zhang et al., 2021) [47]. Additionally, Zhu et al. (2021) examined the response of epidermal proteins in *Spodoptera frugiperda* to various insecticide stresses through combined genomic and transcriptomic analyses [48]. Sun et al. (2021) demonstrated that the overexpression of Cytochrome P450s (CYPs) enhanced insecticide resistance in mosquitoes by integrating transcriptomic and proteomic analyses [49].

In response to insecticide stress, transcription factors serve as crucial regulatory proteins that control the expression of numerous downstream resistance-related functional genes [50,51]. The family of transcription factors involved in regulating detoxification genes in insects include basic helix–loop–helix (bHLH-PAS), nuclear receptors, and basic leucine zipper (bZIP) proteins [52]. These transcription factors transcriptionally regulate the expression of the detoxification of enzyme genes in resistant insects, triggering detoxification and metabolism of insecticides before the chemicals reach the target sites [53,54,55,56].

In this study, based on combined transcriptome and proteome analysis of susceptible and lambda-cyhalothrin-resistant strains of *L. pratensis*, the functional annotation and enrichment analysis of differentially expressed genes and proteins were performed, and the key differentially expressed genes and proteins associated with resistance were identified. Additionally, we also screened transcription factors involved in the regulation of detoxification metabolism, constructed an evolutionary tree of the detoxification gene family, and analyzed detoxification metabolism and related pathways associated with resistance against lambda-cyhalothrin. This study provides a theoretical basis for further research on the molecular mechanism and regulation mechanism underlying pyrethroid resistance in *L. pratensis*.

## 2. Results

### 2.1. Transcriptome Sequencing Results and Analysis

#### 2.1.1. De Novo Assembly and Correlation Analysis

Total RNA was extracted, and cDNA libraries were constructed from two resistant strains (R6, R14) and one susceptible strain (S) of *L. pratensis* and then sequenced on IlluminaHiSeq™2000. After quality control of raw data, filtering low-quality data, checking sequence error rate and GC content distribution, 0.473 Gb clean reads were obtained for subsequent analysis, and the GC content of the sequence data ranged from 46.34% to 46.92%, with the proportion of Q20 and Q30 being 98.8% and 96.3%, respectively (Appendix A). A total of 82,919 unigenes were obtained after assembling the high-quality sequences using Trinity software, with an average length of 880 bp and an N50 length of 1655 (Appendix A). The total number of unigenes with a length distribution between 200 and 300 bp was the highest (Appendix A), indicating that the sequencing assembly results were accurate and reliable and can be used for further research and analysis.

#### 2.1.2. Transcriptome Principal Component Analysis (PCA)

The PCA based on transcriptome sequencing data showed that the sequencing results of R6 and R14 strains had good repeatability and obvious separation trends among different samples (Figure 1); PC1 accounted for the largest variance in the transcriptomic data, representing 67.5% of the total variance. On the PC1 axis, the S, R6, and R14 strains exhibited a clear separation trend, indicating significant differences in gene expression levels. PC2 accounted for the second-largest variance in the data, representing 21% of the total variance. It further distinguished the differences between R6 and R14. On the PC2 axis, the samples of R6 and R14 also showed good reproducibility, indicating that they shared some similarities in transcriptional levels but also exhibited unique gene expression patterns. The PCA demonstrated there were significant differences at the transcriptional levels among the S, R6, and R14 strains.

#### 2.1.3. Identification and Functional Annotation of Transcripts

After aligning 82,919 transcript sequences with GO, KEGG, KOG, Swiss-Prot, and Nr databases, a total of 24,859 annotated transcript sequences were obtained (Appendix A; Appendix A). Using Blastx, the unigene sequences were aligned with the Nr database by Blastx, and the best-matching sequences were selected as the corresponding homologous sequence so as to determine the species to which the homologous sequences belonged. A total of 24,687 unigenes (29.8%) were annotated in the Nr database. Among them, the number of homologous sequences matched with *Cimex lectularius* (53.2%, 7233) is the highest, followed by *Halyomorpha halys* (20.4%, 22,783) and *Lasius niger* (8.1%, 1103). The number of homologous sequences matched with other species, such as *Anoplophora glabripennis*, *Priapulus caudatus*, *Diaphorina citri*, *Exaiptasia castaneum*, *Tribolium castaneum*, and so on, are all below 500 (Appendix A). Appendix A further shows the distribution of homologous sequences of these species, indicating that *L. pratensis* has a closer evolutionary relationship with *Cimex lectularius* and *Halyomorpha halys*.

#### 2.1.4. Analysis of Differentially Expressed Genes (DEGs) in the Transcriptome

During the detection process of differentially expressed transcripts (DEGs), RPKM is used as an indicator to measure the expression level of transcripts, with log2|(Fold Change)| ≥ 1 and FDR < 0.05 as the screening criteria. A total of 16,827 DEGs were identified in S-VS-R6, including 3565 up-regulated DEGs and 13,262 down-regulated DEGs; in S-VS-R14, a total of 27,260 DEGs were identified, including 6538 up-regulated and 20,722 down-regulated DEGs; and in R14-vs-R6, a total of 14,543 DEGs were identified, including 3565 up-regulated DEGs and 11,408 down-regulated DEGs (Figure 2A). The number of differential transcripts in R14 is higher than in R6 (Figure 2B).

#### 2.1.5. GO Enrichment Analysis of the Transcriptome

The results of GO functional annotation indicated (Appendix A) that there were 54, 53, and 43 differentially expressed transcripts annotated in the three sets of DEGs, respectively. Among them, most transcripts were annotated to the “Biological Process” category, including “Metabolic process”, “Cellular process”, and “Single-organism process”. In the “Cellular Component” category, most transcripts are annotated to “Cell”, “Cell part”, “Macromolecular complex”, “Membrane”, “Membrane part”, “Organelle”, and “Organelle part”. In the “Molecular Function” category, the most differentially expressed transcripts are annotated to “Catalytic activity” and “Binding” [57].

#### 2.1.6. KEGG Enrichment Analysis of the Transcriptome

The results of KEGG enrichment analysis indicated (Appendix A) the following: In S-VS-R6, there were 1846 DEGs annotated and enriched in 214 metabolic pathways, with the top 20 mainly enriched in pathways such as “Cell cycle”, “DNA replication”, “Oocytes meiosis”, “Progesterone-mediated oocyte maturation”, and “Adrenergic signaling in cardiomyocytes” (Appendix A). In S-VS-R14, there were 2627 DEGs annotated and enriched in 221 metabolic pathways, with the top 20 mainly enriched in pathways such as “DNA replication”, “Mismatch repair”, “Nucleotide excision repair”, “Ribosome biogenesis in eukaryotes”, and “Base excision repair” (Appendix A). In R14-VS-R6, there were 687 DEGs annotated and enriched in 181 metabolic pathways, with the top 20 pathways enriched in “Neuroactive ligand–receptor interaction”, “Arginine and proline metabolism”, “Glycine, serine and threonine metabolism”, “Nitrogen metabolism”, “Glycerolipid metabolism”, and “Glutathione metabolism” (Appendix A).

#### 2.1.7. Identification and Phylogenetic Analysis of Detoxification Metabolism-Related Genes

A total of 119 P450 genes, 39 GST genes, 45 CarE genes, and 123 ABC genes were identified in the *L. pratensis* transcriptome. After excluding shorter sequences and those with greater genetic distances, the remaining sequences have complete open reading frames (ORFs). The average length of the Cytochrome P450, GST, GarE, and ABC transporter gene sequences are 1125, 558, 1097, and 2043 bp, respectively, encoding 306–375, 135–169, 135–169, and 446–857 amino acids. The genes annotated with Cytochrome P450, GST, and ABC were analyzed using EdiSeq software (Version 8.1) to match amino acid sequences and aligned with the Cytochrome P450, Glutathione-S-transferase, Carboxylesterase, and ABC transporter amino acid sequences of *Cimex lectularius*, *Apolygus lucorum*, *Halyomorpha halys*, *Nesidiocoris tenuis*, *Subpsaltria yangi*, *Macrosteles quadrilineatus*, and *Laodelphax striatellus* in the NCBI database to construct phylogenetic trees. P450 gene sequences were divided into three evolutionary branches belonging to the *CYP3* (6 genes), *CYP4* (11 genes), and *CYP6* subfamily (24 genes) (Figure 3). There were 23 GST genes distributed across five different subfamilies: *GST-Delta (D)*, *GST-Epsilon (E)*, *GST-Zeta (Z)*, *GST-Theta (T)*, and *GST-Sigma (S)* (Figure 4); 27 ABC genes were distributed across five different subfamilies: *ABC-A*, *ABC-B*, *ABC-C*, *ABC-F*, and *ABC-G* (Figure 5); and 15 CarE genes were distributed into two branches: Acetylcholinesterase and α/β hydrolases (Figure 6).

#### 2.1.8. Differential Expression Analysis of Detoxification Metabolism-Related Genes

Based on the Nr annotation results of differentially expressed transcripts, we dug up and analyzed genes associated with the detoxification metabolism. Genes related to resistance were screened with the criteria of Fold Change (FC) ≥ 2 and FDR < 0.05. In S-VS-R6, 26 genes annotated as Cytochrome P450 were selected, with 10 significantly up-regulated and 16 significantly down-regulated; 43 genes annotated as ABC transporters were selected, with 4 significantly up-regulated and 39 significantly down-regulated; and 8 genes annotated as GST were selected, with 6 significantly up-regulated and 2 significantly down-regulated. In S-VS-R14, 36 Cytochrome P450 genes were selected, with 13 significantly up-regulated and 23 significantly down-regulated; 59 ABC transporter genes were selected, with 7 significantly up-regulated and 52 significantly down-regulated; 8 GST genes were selected, with 2 significantly up-regulated and 6 significantly down-regulated. Figure 7 shows that the differential expression of detoxification metabolism genes is related to resistance from different *L. pratensis* strains.

#### 2.1.9. Analysis of Transcription Factor (TF) Families in *L. pratensis*

Based on the assembled unigenes of *L. pratensis*, a preliminary selection of 1499 transcription factors (TFs) belonging to 64 families was carried out, including 956 TFs from the zf-C2H2 family, 70 TFs from the Homeobox family, 50 TFs from the THAP family, 43 TFs from the MYB family, 42 TFs from the bHLH family, 39 TFs from the HTH family, 32 TFs from the HMG family, 21 TFs from the TF_bZIP family, 19 TFs from the Fork head family, and 12 TFs from the zf-LITAF-like family (Table 1; Appendix A). In S-VS-R6, a total of 269 differential transcription factors were identified, with 184 up-regulated and 85 down-regulated; in S-VS-R14, a total of 744 differential transcription factors were identified, with 698 up-regulated and 146 down-regulated. The identified differential transcription factor families are mainly distributed in the zf-C2H2, Homeobox, THAP, MYB, bHLH, HTH, HMG, and TF_bZIP families (Table 2).

#### 2.1.10. Analysis of Transcription Factors Related to Lambda-Cyhalothrin Resistance in *L. pratensis*

Based on a literature search (Gao et al., 2018 [58]) and transcription factor annotation information, the transcription factor families related to resistance in S-VS-R6 and S-VS-R14 were identified as zf-C2H2, TF_bZIP, bHLH, SF-like, C_EBP, RXR-like, zf-GATA, ZBTB Homeobox, Fork head, etc. (Appendix A). The heat map (Figure 8) shows that the expression levels of zf-C2H2, TF_bZIP, bHLH, SF-like, C_EBP, and RXR-like transcription factors in R6 and R14 are significantly higher than in S, indicating their important role in the response to external stress. At the same time, we also mined key transcription factors that respond to the development of pyrethroid resistance, mainly concentrated in the TF bZIP, zf-GATA, ZBTB, SF-like, Homeobox, Fork head families (Table 3), among which CncC/Maf, CREB, GATA, FOXO, etc., are the transcription factors involved in the metabolic pathways of pyrethroid insecticides and the focus of subsequent regulatory mechanism research.

#### 2.1.11. RT-qPCR Validation Results

Based on the DEG result analysis, 10 differentially expressed genes and 10 differentially expressed transcription factors were selected for RT-qPCR validation. The results indicated that the changes in the fluorescence quantitative expression levels of DEGs by RT-qPCR and RNA-Seq were consistent with the gene expression abundance (Figure 9), which suggested that the transcriptome sequencing results were reliable and can be used for further research. To ensure the accuracy and reliability of the qPCR results, we ran standard curves for each of the 10 selected genes and calculated the amplification efficiency for each gene. Appendix A lists the slopes of the standard curves and the calculated amplification efficiencies for each gene. The amplification efficiencies of all genes were within the range of 98–102%, meeting the requirements of the 2^−ΔΔCt^ method.

### 2.2. Proteome Sequencing Results and Analysis

#### 2.2.1. Proteomics Quality Control Assessment and Quantitative Analysis

The iTraq/TMT technology (AB Sciex, Framingham, MA, USA) was used for absolute and relative quantification of proteins, which was analyzed using Mascot 2.3.02 software (Matrix Science, London, UK). The total number of secondary spectra identified in this experiment was 360,572, and the number of matching spectra in the database was 26,324. The total number of peptides identified was 22,935, the number of peptides was 5230, the number of unique peptide sequences identified was 4943, and the total number of proteins identified was 1664 (Appendix A).

The molecular weight distribution of proteins is an important index for identifying the size of proteins, and the results showed that the proteins were effectively separated in the molecular weight range of 15~220 kDa (Appendix A), and the total amount and quality of proteins fulfilled the experiment requirements. After that, quality control of proteomic data was carried out. The results showed that the peptides were mainly distributed in the length range of 7–20 bp (Appendix A), and the distribution of protein molecular weight and coverage was normal (Appendix A), which indicated that the proteolytic digestion effect was satisfactory, and mass spectrometry identification and data retrieval met the requirements of follow-up analysis.

#### 2.2.2. KOG Annotation Analysis of the Proteome

The expressed proteins of the susceptible (S) and resistant (R14) strains of *L. pratensis* were compared with the KOG database. The annotation results indicated that the main enrichments were found in the following 10 major KOG categories: “General function prediction only”, “Signal transduction mechanisms”, “Energy production and conversion”, “Lipid transport and metabolism”, “Translation, ribosomal structure and biogenesis”, “Defense mechanisms”, “RNA processing and modification”, “Cytoskeleton”, and “Amino acid transport and metabolism” (Appendix A Appendix A).

#### 2.2.3. Principal Component Analysis of Proteome

The principal component analysis (PCA) of six samples from two groups of *L. pratensis* was performed using the R package (https://ggplot2.tidyverse.org/, accessed on 30 February 2022) (R Foundation for Statistical Computing, Vienna, Austria). As shown in Figure 10, PC1 accounted for the largest variance in the proteomic data, representing 59.8% of the total variance. It primarily reflected the differences between the susceptible strain (S) and the 14th-generation resistant strain (R14). On the PC1 axis, the S and R14 strains exhibited a complete separation trend, indicating significant differences in protein expression levels. PC2 accounted for the second-largest variance in the data, representing 13.1% of the total variance, and further distinguished the differences among the samples within the R14 strain. On the PC2 axis, the three replicates clustered closely together, indicating good reproducibility of the PCA results. The PCA results demonstrated significant differences between S and the R14 strain in terms of gene expression and protein expression levels. These differences are likely closely related to the development of lambda-cyhalothrin resistance. Additionally, the R14 strain exhibited good reproducibility at both the transcriptional and translational levels, further validating the reliability and consistency of data.

#### 2.2.4. Analysis of Differentially Expressed Proteins (DEPs)

For the identification, a threshold was set where a protein with a Fold Change (FC) ≥ 1.2 and a *p*-value < 0.05 was considered to be differentially expressed. After analyzing the proteins across six samples, a total of 178 DEPs were identified. The results showed (Figure 11A,C) that 11 proteins were significantly up-regulated, and 20 proteins were significantly down-regulated in the group of R14-1-VS-R14-2, while 23 proteins were significantly up-regulated, and 31 proteins were significantly down-regulated in the R14-1-VS-R14-3 group. For R14-1-VS-S-1, 55 proteins were significantly up-regulated, and 65 proteins were significantly down-regulated. For R14-1-VS-S-2, 62 proteins were significantly up-regulated, and 67 proteins were significantly down-regulated, while in the group of R14-1-VS-S-3, 72 proteins were significantly up-regulated, and 69 proteins were significantly down-regulated. In summary, in “S-VS-R14”, 85 proteins were significantly up-regulated, and 93 proteins were significantly down-regulated (Figure 11B).

#### 2.2.5. GO Enrichment Analysis of Differentially Expressed Proteins (DEPs)

The GO functional annotation results indicated that 146 DEPs were annotated to different GO terms. Among them, in the “Biological Process” category, the most annotated transcripts were related to “Cellular processes”, “Metabolic process”, and “Single-organism process”. In the “Cellular Component” category, most transcripts were found in the subcategories of “Cell”, “Cell part”, “Macromolecular complex”, “Membrane”, “Organelle”, and “Organelle part”. The subcategories with the most differentially expressed transcripts in the “Molecular Function” category is “Catalytic activity” and “Binding” (Appendix A).

#### 2.2.6. KEGG Analysis of Differentially Expressed Proteins (DEPs)

In “R14-VS-S-1”, 41 pathways were enriched, with the most significant pathways bei ng “Neuroactive ligand–receptor interaction”, “Glycosphingolipid biosynthesis–globo series”, “Oxidative phosphorylation”, and “Phagosome”. In “R14-VS-S-2”, 46 pathways were enriched, with the most significant pathway being “Phagosome”, and also enriched in “Metabolism of xenobiotics by Cytochrome P450” and “Drug metabolism–Cytochrome P450”. In “R14-VS-S-3”, 46 pathways were enriched, with the most significant pathways being “Ribosome”, “Phagosome”, “Lysosome”, and “Glycosphingolipid biosynthesis–globo series”, and also enriched in “Metabolism of xenobiotics by Cytochrome P450” and “Drug metabolism–Cytochrome P450” (Appendix A).

#### 2.2.7. Systematic Analysis of DEPs Related to Lambda-Cyhalothrin Resistance

Screening for resistance-related differential proteins was conducted using the criteria of Fold Change (FC) ≥ 1.2 and FDR (false discovery rate) < 0.05. A heat map was plotted based on the RPKM values of differential proteins (Figure 12). In S-VS-R14, three Cytochrome P450 proteins and two Glutathione S-transferases were selected. Among them, a gene annotated as Cytochrome P450 *CYP6A13* was significantly up-regulated.

### 2.3. Correlation Analysis of DEGs and DEPs

Transcriptomic and proteomic data were integrated and analyzed to compare the transcriptional and translation levels of lambda-cyhalothrin resistance of *L. pratensis* and further explore the drug-related proteins and coding genes. The Venn diagram showed that the detected DEGs/DEPs (*p*-value < 0.05, FC > 1.2) were 37,007 and 178, respectively, with 140 differential genes and proteins showing consistent expression trends (Figure 13). The gene expression changes in the transcriptome and proteome were categorized. The diagram was divided into nine quadrants by the dashed lines of the horizontal and vertical coordinates (Figure 14), with the dashed line on the horizontal axis representing the Fold Change threshold of transcriptome and the dashed line on the vertical axis representing the Fold Change threshold of proteome.

### 2.4. Metabolic Pathway Analysis of Transcriptomic and Proteomic Data

The GO and KEGG enrichment analyses were performed for DEGs and DEPs with consistent expression trends in quadrants 3 and 7 of Figure 15. In quadrant 3, the DEGs/DEPs with the same expression trend were co-up-regulated, and the GO enrichment loop shows that DEGs/DEPs are mainly enriched in “Molecular Information” (Figure 16A,C). The KEGG pathway enrichment analysis indicates that 23 DEGs/DEPs genes are enriched in 23 pathways, which are primarily enriched in pathways such as “Sphingolipid metabolism”, “Glycosphingolipid biosynthesis–globo and isoglobo series”, “Dorso–ventral axis formation”, “Ribosome”, “Terpenoid backbone biosynthesis”, “ABC transporters”, “RNA transport” and “Peroxisome” (Figure 17A,C). Quadrant 7 contains DEGs/DEPs that are consistently down-regulated, and the GO enrichment loop shows that DEGs/DEPs are mainly enriched in “Biological Process” (Figure 16B,D). Based on the KEGG database, 17 DEGs/DEPs genes are enriched in 14 pathways. They are primarily enriched in pathways such as “Oxidative phosphorylation”, “Phagosome”, “Fatty acid biosynthesis”, “Selenocompound metabolism”, “Protein processing in endoplasmic reticulum”, “Biosynthesis of amino acids”, “Neuroactive ligand–receptor interaction”, “Peroxisome”, and “Carbon metabolism” (Figure 17B,D). When combining with key genes identified at the transcriptional level in previous studies, it was found in the correlation analysis that one P450 gene is significantly up-regulated at both the transcriptional and translational levels, annotated as Cytochrome P450 *CYP6A13* (GenBank accession number: MN782520) [59] suggesting that its overexpression might be involved in resistance to lambda-cyhalothrin.

## 3. Discussion

Insects develop resistance to exogenous insecticides primarily through enhanced detoxification metabolism [60,61] and the production of target site mutations [59,62]. The enhancement of insecticide detoxification metabolic activity mediated by Cytochrome P450 is one of the main mechanisms of insecticide resistance [63]. In this study, under the selective pressure of lambda-cyhalothrin, various detoxification enzyme genes were significantly up-regulated, indicating their potential involvement in the detoxification metabolism of permethrin. For instance, the *ABCG4* gene expression in the larvae of *Subgenus stephensi* was induced significantly up-regulated under the LD_50_ of cypermethrin [64]. Hu et al. (2019) found that three GST genes were highly expressed in cypermethrin-resistant populations of *Spodoptera exigua* [65]. Based on the transcriptome data of *L. pratensis*, these detoxification metabolism genes related to resistance were identified and subjected to phylogenetic analysis: CYPs were mainly found in the CYP3, CYP4, and CYP6 evolutionary branches. Cytochrome P450 enzymes in insects are currently divided into six evolutionary branches, namely CYP2, CYP3, CYP4, CYP16, CYP20, and mitochondrial CYP evolutionary branches [66,67]. Among them, the CYP3 evolutionary branch can be further divided into CYP6 and CYP9 sub-branches, with genes related to resistance mainly concentrated in CYP3 and CYP4 [68], and CYP6 family genes are closely related to the metabolism of plant secondary substances [69], indicating that these CYPs identified in *L. pratensis* play a significant role in resistance to lambda-cyhalothrin.

Insects’ GST genes are divided into six subfamilies: Delta, Epsilon, Omega, Sigma, Theta, and Zeta [70,71]. In our study, 22 GST genes were successfully screened from the transcriptome database of *L. pratensis*, which are distributed across five different evolutionary branches: Delta, Epsilon, Zeta, Theta, and Sigma. Within the GST superfamily, the Epsilon and Delta classes are two subfamilies unique to insects, capable of degrading a variety of insecticides, including Organophosphates, Organochlorines, and pyrethroids [72,73], indicating that GSTs may be involved in the detoxification metabolism process of *L. pratensis* to lambda-cyhalothrin. CarE is one of the three major detoxification enzymes in insects, which can hydrolyze exogenous substances into corresponding alcohols or acids to participate in the metabolism of toxic substances [74]. Shui et al. (2024) have proven that the Carboxylesterase *EoCarE592* of *Ectropis oblique hypulina Wehrli* can metabolize insecticides such as lambda-cyhalothrin [75]. The 15 CarE genes identified in this study are distributed within the Acetylcholinesterase subfamily, suggesting that they may play a key role in the detoxification process of *L. pratensis*. Insect ABC transporters are divided into eight subfamilies (*ABCA*-*ABCH*) [76], and various ABC transporter families play an important role in the degradation process of endogenous and exogenous toxins in organisms [77]. The *ABCG3* gene expression in *Bemisia tabaci* was up-regulated by 3.3 times after treatment with imidacloprid [78], and the *ABCB2* gene expression in *Sogatella furcifera* showed significant changes after treatment with different concentrations of buprofezin, thiamethoxam, and abamectin [79]. In this study, 27 differentially expressed ABC genes were screened from the transcriptome of *L. pratensis*, which are distributed in five different subfamilies—*ABC-A*, *ABC-B*, *ABC-C*, *ABC-F*, and *ABC-G*—suggesting that ABC transporters play a significant role in the metabolism of exogenous compounds in *L. pratensis*.

The frequent use of insecticides inevitably causes continuous increasing resistance among insects [80,81,82]. In addition to gene duplication or gene amplification, the overexpression of detoxification enzyme genes is an important mechanism for resistance development in insects [83,84]. Transcription factors play a large role in the hosts’ response to external stress [85]. C2H2 zinc finger proteins, as a large class of transcription factors and nuclear receptor proteins, regulate the expression of target genes (enhancement or inhibition) by binding to promoters [86,87]. The C2H2 zinc finger structural domain is the most abundant class of transcription factors in eukaryotes [88,89], and proteins containing the C2H2 motif account for about 2% of all genes in *Drosophila* [90]. Studies have shown that C2H2 zinc fingers can respond to stress from pathogens or xenobiotics. For example, Lv et al. (2023) have shown that the C2H2 zinc finger transcription factor *CF2-II* can bind to the promoters of multiple resistance-related genes through dual-luciferase reporter gene assays, yeast one-hybrid systems, and electrophoretic mobility shift assays, thereby regulating the ABC transporter gene expression of the spiromesifen-resistant *Aphis gossypii* [91]. Baculoviruses are highly pathogenic to arthropods [92], and the expression of the C2H2 gene *BmZAD69* in *Bombyx mori* can respond to the infection of nuclear polyhedrosis virus (BmNPV) [93]. C2H2 zinc finger proteins can also regulate the immune homeostasis of insects. The *Drosophila* C2H2 zinc finger protein *IMZF* can suppress the immune response of the IMD (immune deficiency) signaling pathway [94], preventing the imbalance of immune intensity and duration from having adverse effects on the organism [95]. In this study, the number of transcription factors belonging to the C2H2-ZF structural domain family accounts for 60% of the total number of differential transcription factors, indicating that under the stress of lambda-cyhalothrin, transcription factors containing the C2H2-ZF structural domain play a significant role in the transcriptional regulation of the detoxification of enzyme genes and in maintaining the homeostasis of an organism. bZIP proteins and Nf2 transcription factors play important roles together in the growth and development of insects and in resisting insecticides [96,97,98,99], among which CncC (cap ’n’ collar isoform-C) is involved in the transcriptional regulation of insect detoxification metabolism against insecticides [100,101,102]. Under normal conditions, CncC binds to Keap1 in the cytoplasm. When subjected to stress from xenobiotics or endogenous peroxides, CncC dissociates from Keap1, translocates to the nucleus, and forms a heterodimer with Maf. The heterodimer binds to specific antioxidant response elements (AREs) upstream of genes [103] and then triggers the transcriptional regulation of detoxification enzyme (P450s, GSTs) genes. Studies have shown that exposure to pyrethroid insecticides leads to oxidative stress [104], and the insecticide-activated ROS-CncC pathway mediates the xenobiotic detoxification pathway to enhance the insects’ tolerance to chemicals [105]. The bZIP transcription factor CREB enhances the resistance of *Bemisia tabaci* to neonicotinoid insecticides by binding to the CYP6CM1 promoter [106]. In this study, several bZIP differentially expressed transcription factors were screened, indicating that lambda-cyhalothrin induced oxidative stress responses in *L. pratensis* and activated the up-regulated expression of related bZIP family transcription factors, thereby transcriptionally regulating detoxification enzyme metabolism of insecticides and antioxidant responses. The bHLH proteins are widely present in insects and play an important regulatory role in their growth and development [107]. The bHLH-PAS family transcription factor *TcMet* can regulate the synthesis of the juvenile hormone (juvenile hormone, JH) during the larval stage, and RNAi experiments have proven that Met plays a key role in preventing the premature development of adult structures during the larval–pupal process in *Tribolium castaneum* [108]. In this study, the differentially expressed transcription factors of the bHLH family play an important role in maintaining the normal growth and development of insects under the selection pressure of lambda-cyhalothrin, all of which were identified.

A combined enrichment analysis of DEGs and DEPs with the same expression trend was performed. The results indicated that the up-regulated DEGs and DEPs were significantly enriched in multiple detoxification metabolic pathways, especially in pathways such as sphingolipid metabolism, Glycosphingolipid biosynthesis, Ribosomes, Terpenoid backbone biosynthesis, ABC transporters, RNA transport, and Peroxisomes. Sphingolipids are biologically active lipids found in cell membranes and major membrane sphingolipids, such as sphingomyelin (SM) and Glycosphingolipids (GSL), which are core components of specific membrane microdomains [109]. They can regulate the biophysical properties of biological membranes. Some researchers reported that sphingolipids can control key cellular functions [110], such as cell cycle, aging, apoptosis, cell migration, and inflammation [111]. The up-regulated differential genes and proteins enriched in the “Sphingolipid metabolism” and “Glycosphingolipid biosynthesis” pathways indicated that the plant bug can regulate their key cellular functions, including apoptosis, cell cycle, and aging, when exposed to deltamethrin. Apoptosis is inextricably linked to oxidative stress. Reactive oxygen species (ROS) in cells can lead to apoptosis [112], directly damaging cellular macromolecules such as proteins, lipids, and nucleic acids and causing cell death or apoptosis [113]. As the direct function executor of living organisms, the functional changes in some detoxification metabolism proteins are key factors in producing resistance [114]. For insects, when subjected to external stimuli, the conformation of their proteins and enzymes can be easily disrupted, leading to the rapid production of reactive oxygen species (ROS) within the organism [115]. Pyrethroid insecticides can cause oxidative stress reactions in various organisms, including insects and mammals [116,117,118]. The differentially expressed genes and proteins in S-VS-R6 are mainly enriched in pathways such as “cell cycle”, “DNA replication”, “Mismatch repair”, “Nucleotide excision repair”, and “Base excision repair”, indicating that lambda-cyhalothrin-induced oxidative stress may lead to oxidative damage to nuclear acids and activate DNA damage repair mechanisms. Enrichment in the “Peroxisome” pathway suggests that *L. pratensis* initiates its antioxidant enzyme system in response to oxidative stress caused by exogenous insecticides in order to maintain the redox balance within the cell. Antioxidant enzymes such as superoxide dismutase, catalase, and thioredoxin work together against oxidative damage [119]. Additionally, the co-up-regulated DEGs/DEPs were also enriched in the “Terpenoid backbone biosynthesis” pathway. In insects, the juvenile hormone (JH) maintains the growth and development of larvae. A higher titer in the larval stage can maintain the larval state, prepare for the vitellogenic stage, promote the maturation and growth of primary follicles, and prevent metamorphosis. In contrast, the 20-hydroxyecdysone (20E) antagonizes JH and promotes metamorphic development [120,121]. The antagonistic interaction between JH and 20E ensures precise regulation of the metamorphosis process [85]. The Terpenoid backbone is the biosynthetic precursor for insect JH and 20E. Transcriptome KEGG pathway analysis shows that DEGs are significantly down-regulated in the JH synthesis pathway and significantly up-regulated in the 20E synthesis pathway, which is consistent with the report of Jiang et al. (2023), in which it was found that the beetles *Tribolium castaneum* improved resistance by reducing fertility after being induced by deltamethrin [122]. Organisms consume a large amount of energy when detoxifying and reproducing (Yang et al., 2020). In the “Terpenoid backbone biosynthesis” and “insect hormone synthesis” pathways, the down-regulation of the JH synthesis pathway and the up-regulation of the 20E synthesis pathway indicated that *L. pratensis* also increased its resistance to insecticides by reducing fertility [123].

The co-down-regulated DEGs/DEPs are mainly enriched in pathways related to growth and development regulation, such as Oxidative phosphorylation, Fatty acid biosynthesis, Terpenoid backbone biosynthesis, and Neuroactive ligand–receptor interaction. The Oxidative phosphorylation system is closely related to energy generation [124]. Ren et al. demonstrated that the expression of genes in the mitochondrial electron transport chain reduced on *Bombyx mori* after they were treated with lambda-cyhalothrin, thereby inhibiting the Oxidative phosphorylation pathway and decreasing ATP synthesis [125]. Gao et al. reported that the genes on the mitochondrial respiratory chain pathway of *Drosophila melanogaster* were generally down-regulated after the application of six different insecticides, including chlorpyrifos, cypermethrin, and imidacloprid at their LC_10_ concentrations [58,126]. Similarly, exposure to the LC_10_ concentration of the aforementioned five insecticides showed a similar gene downtrend on *Plutella xylostella*, which is consistent with the results of our study, indicating that lambda-cyhalothrin can affect the energy metabolism of *L. pratensis* by inhibiting the Oxidative phosphorylation pathway. The co-down-regulated DEGs/DEPs are also enriched in the “Neuroactive ligand–receptor interaction” pathway, which involves various neurotransmitters and their receptors. Through these interactions, downstream signaling pathways are activated or inhibited, thereby regulating biological neural activity and behavioral performance [127]. Pyrethroid insecticides typically act on the nervous system of insects, causing disruption of the nervous system, rapid onset of poisoning symptoms, loss of coordinated movement, and manifesting as continuous twitching, ultimately leading to death [128,129]. Voltage-gated sodium channels (VGSCs) are the targets of pyrethroid insecticides [130]. When the drug is combined with the voltage-gated sodium channel, it changes the opening time of the ion channel, leading to excessive excitation of the nervous system and death of the target organism [131]. The significant down-regulation of differential genes and proteins in this pathway clearly illustrates the action mechanism of pyrethroid insecticides. Fatty acids are essential for maintaining the normal growth, development, and physiological activities of insects. Fatty acid biosynthesis is a complex multi-step reaction process, primarily carried out by enzymes such as acetyl–coenzyme A carboxylase (ACC), Fatty acid synthase (FAS), elongase of long-chain Fatty acid (ELO), desaturase, Fatty acid desaturase (FAD), and fatty acyl-CoA reductase (FAR) [132]. ACC and ELO can affect insect reproductive capacity. Mutation in ACC^CG11198^ leads to embryo death of *Drosophila melanogaster* [133], and interference with ACC reduces the egg-laying and reproductive capacity of *Aedes aegypti* [134]. Additionally, RNA interference in the ELO^CG3971^ target gene in the late stages of spermatocyte development in the test fruit flies *Drosophila* also causes male sterility [135]. Insects can improve their metabolic detoxification effects against insecticides by reducing fertility [122], which down-regulated the expression of DEGs/DEPs and enriched the “Fatty acid synthesis” and “Fatty acid metabolism” pathways, indicating that lambda-cyhalothrin affects normal reproduction and development of *L. pratensis*, which was inferred that *L. pratensis* might adopt an energy allocation strategy in the process of resistance formation, giving priority to the detoxification processes rather than reproduction. It is of great significance for understanding the environmental adaptability of pests and providing a reference for the development of pest management strategies on alfalfa.

## 4. Materials and Methods

### 4.1. Collection and Rearing of Test Insects

The initial colony of *L. pratensis*, consisting of approximately 1000 mixed nymphs and adults, was collected via net-sweeping from alfalfa (ZhongCao No. 3) fields at the Helin County Experimental Station of the Institute of Grassland Research (40°60′ N, 111°80′ E), Chinese Academy of Agricultural Sciences (Hohhot, Inner Mongolia, China) in April 2014. The experimental station was a breeding base for alfalfa and had never sprayed pesticides in history. The original population had not been exposed to any chemical insecticides and was invariably reared in the laboratory. The LD_50_ value was 9.33 ng a.i./adult, and it was used as a susceptible strain (S) for future experiments [19]. The resistant strain was originally collected from Helin County Experimental Station of the Institute of Grassland Research in April 2015, maintained in the laboratory and selected using the insecticide film method with lambda-cyhalothrin for more than 20 generations, finally showing higher 90-fold resistance in laboratory tests. The resistant strains were continuously selected for 6 (R6, 9.33 ng a.i./adult, 7.5-fold) and 14 (R14, 55.03 ng a.i./adult, 42.5-fold) generations; the healthy virgin adults were collected as lambda-cyhalothrin-resistant samples used for the downstream experiment [27]. The colonies were maintained in a growth chamber at 25 ± 1 °C and 50 ± 10% relative humidity, with a light cycle of 16 h of light and 8 h of darkness (L16 h: D8 h). Rearing was conducted in aerated plastic boxes (20 cm × 20 cm × 20 cm), with one open side covered by sterile gauze, and the insects were fed on insecticide-free green bean pods (*Phaseolus vulgaris* L.) following the rearing method described by Jia et al. [136].

### 4.2. cDNA Library Construction and Transcriptome Sequencing

The RNA-Seq of the adult *L. pratensis* was conducted by Guangzhou JidiAo Technology Service Co., Ltd. (Guangzhou, China). Transcriptome sequencing included three samples with three biological replicates, and a total of nine cDNA libraries were constructed. Following the extraction of total RNA from the samples, mRNA was enriched using magnetic beads with Oligo (dT). The obtained mRNAs were then divided into fragments randomly with fragmentation buffer, and the mRNA fragments divided into shorts were taken as templates. Subsequently, the first strand of cDNA was synthesized using the fragmented mRNA as a template, along with six-base random primers (random hexamers), buffer, dNTPs, RNase H, and DNA polymerase I for synthesis of the second strand of cDNA. This cDNA was purified using the QiaQuick PCR kit and eluted with EB buffer through end repair, addition of base A, and attachment of sequencing connectors. The target-sized fragments were recovered via agarose gel electrophoresis and amplified through PCR. The constructed library underwent transcriptome sequencing on the Illumina HiSeq™ 2000 high-throughput sequencing platform (Illumina Inc., San Diego, CA, USA). To filter the raw data obtained from sequencing, we used FastQC for quality assessment and Trimmomatic for data filtering. The filtering process ensured that the resulting clean reads met the quality criteria of Q20 and Q30, with the minimum read length set at 200 bp. This stringent filtering produced high-quality, clean reads to ensure the accuracy and reliability of the sequencing data.

### 4.3. Data Assembly and Functional Annotation

The clean reads were assembled and spliced using Trinity software (v2.4.0) to generate transcript sequences, with the assembled reads lacking N-terminal sequences classified as unigene. All assembled unigene sequences were aligned to the protein databases Nr, Swiss-Prot, KOG, and KEGG (e-value < 0.00001) using Blastx. This analysis identified the protein with the highest sequence similarity to each unigene, thereby providing functional annotation information for those unigenes.

### 4.4. Identification of Differentially Expressed Genes (DEGs) and Data Analysis

The RPKM (Reads Per Kilobase Per Million) values of the transcripts were calculated using RSEM (RNA-Seq by Expectation–Maximization.) software (version 1.3.3). Differential expression analysis was conducted using DESeq2 (http://www.bioconductor.org/packages/release/bioc/html/DESeq2.html, accessed on 3 February 2022), employing a Fold Change (FC) threshold of ≥ 2 and a false discovery rate (FDR) of <0.05 as screening criteria for the transcriptome data. Differentially expressed genes were further analyzed through principal component analysis using the R package (https://ggplot2.tidyverse.org/, accessed on 30 February 2022). Additionally, the differentially expressed transcripts were compared with the Gene Ontology (GO) and Kyoto Encyclopedia of Genes and Genomes (KEGG) databases for annotation and enrichment analysis.

### 4.5. Identification and Phylogenetic Analysis of Resistance-Related Genes of L. pratensis

Using sequencing data from the transcriptome of both susceptible and resistant strains of *L. pratensis*, we conducted an analysis for screening resistance-related genes. Genes annotated as Cytochrome P450, Glutathione S-transferase, Carboxylesterase, and ABC transporter proteins were identified based on results from the Non-redundant Protein Library (NR), Swiss-Prot, KEGG, GO, and other databases. Corresponding gene sequences were retrieved from the transcriptome database according to the sequence numbers of the identified genes. The longest open reading frame (ORF) of each unigene was determined using the NCBI ORF Finder tool (http://www.ncbi.nlm.nih.gov/gorf/gorf.html, accessed on 5 March 2022). Subsequently, these sequences were compared using NCBI’s Protein Blast online tool, with Cytochrome P450, GST, and ABC proteins from other insects serving as query sequences. The unigene exhibiting the highest similarity was identified as the homologous sequence based on the comparison’s score and e-value. Additionally, shorter sequences extracted from the transcriptome and those with significant genetic distance were excluded from further analysis. A phylogenetic tree with 1000 bootstrap values was constructed using the neighbor-joining method in MEGA 11.0 software. The evolutionary tree was subsequently optimized through the evolview online platform (https://www.omicsclass.com/article/671, accessed on 30 March 2022), ultimately elucidating the evolutionary relationships among the resistance genes in *L. pratensis*.

### 4.6. Screening for Resistance-Related Transcription Factors

Based on the transcriptome analysis of *L. pratensis*, the assembled transcript sequences are compared and annotated with public databases to perform gene functional annotation. The CDS sequences obtained from Blast alignment are translated into amino acid sequences. The amino acid sequences are then compared with the AnimalTFDB3.0 database (https://doi.org/10.1093/nar/gky822, accessed on 5 May 2024) to identify transcription factor families [137]. Differentially expressed transcription factors are screened with the criteria of Fold Change (FC) ≥ 2 and FDR < 0.05.

### 4.7. Real-Time Fluorescence Quantitative PCR (RT-qPCR) Validation

To assess the reliability of the transcriptome data, 10 differentially expressed genes (DEGs) and transcription factors were randomly selected for validation using real-time fluorescence quantitative PCR (RT-qPCR). Meanwhile, total RNA was extracted from samples S, R6, and R14 using the TaKara RNA kit (Takara Biomedical Technology Co., Ltd., Beijing, China). First-strand reverse transcription was conducted using the TaKara Reverse Transcription Kit (Takara Biomedical Technology Co., Ltd., Beijing, China). The total reaction for RT-qPCR comprised 10 μL mixture, consisting of 5 μL of GoTaq qPCR Master Mix (fluorescent dye), 3.6 μL of nuclease-free water, 0.2 μL of each forward and reverse primer, and 1 μL of cDNA sample. The PCR reaction conditions included an initial pre-denaturation step at 95 °C for 10 min, followed by 40 cycles of denaturation at 95 °C for 15 s, and annealing at 60 °C for 1 min. Subsequently, melting curve analysis was performed at 94 °C for 30 s, 60 °C for 90 s°C, and 94 °C for 10 s to confirm primer specificity. Each treatment was replicated three times biologically, and three technical replicates were conducted for each group. The primer sequences for the target and internal reference genes [138] were designed using Primer Premier 6.0 software (Appendix A), and gene expression levels were quantified using the 2^−ΔΔCt^ method [139,140].

### 4.8. Total Protein Extraction

Samples were collected from the resistant (R14) strain and the susceptible (S) strain, with three biological replicates selected for each group. The samples were placed in freezing tubes, immediately snap-frozen in liquid nitrogen, and subsequently transferred to a −80 °C freezer for protein extraction. Total protein extraction from *L. pratensis* was performed using the cold acetone method. The samples were ground to powders in liquid nitrogen and then dissolved in 2 mL of lysis buffer consisting of 8 M urea, 2% SDS, and 1× Protease Inhibitor Cocktail (Roche Ltd., Basel, Switzerland). The supernatant was sonicated on ice for 30 min, centrifuged at 4360× *g* at 4 °C for 30 min, and transferred to a new tube. For each sample, ice-cold acetone was added and incubated at −20 °C overnight. The precipitates were washed three times with acetone and dissolved in 8 M urea through sonication on ice, and the protein concentration was subsequently assessed using SDS-PAGE.

### 4.9. Enzymatic Digestion and ITRAQ Labeling

In this study, 100 μg of protein was transferred to a new tube, and 8 M urea was added, adjusting the final volume to 100 μL by including 1 M DTT (DL-dithiothreitol). The mixture was incubated at 37 °C for 1 h. Subsequently, an additional 120 μL of the sample was treated with 55 mM iodoacetamide and incubated at room temperature for 20 min, protected from light. Proteins were then precipitated using frozen acetone and re-dissolved in 100 μL of TEAB. Tryptic digestion was conducted overnight at 37 °C using serial-grade modified trypsin (Promega, Madison, WI, USA). The resulting peptides were labeled using the isobaric tags for relative and absolute quantification (iTRAQ) technique. The labeled samples were combined, dried under vacuum, and stored at −20 °C for subsequent mass spectrometry analysis.

### 4.10. Liquid Chromatography–Mass Spectrometry (LC-MS) Analysis

The peptide mixture was re-dissolved in buffer A (buffer A: 20 mM ammonium formate in water, pH 10.0, adjusted with ammonium hydroxide) and subjected to high pH separation using an Ultimate 3000 system (Thermo Fisher Scientific, Waltham, MA, USA), connected to a reversed-phase column [XBridge C18 column, 4.6 mm × 250 mm, 5 μm, (Waters Corporation, Milford, MA, USA)]. A linear gradient from 5% B to 45% B was applied to perform high pH separation in 40 min (B: 20 mM ammonium formate, 80% ACN, pH 10.0, adjusted with ammonium hydroxide). The column was re-equilibrated under initial conditions for 15 min, with a column flow rate of 1 mL/min and a column temperature of 30 °C. Twelve fractions were collected, and each fraction was dried in a vacuum concentrator for further analysis. Peptides were re-suspended in 30 μL of solvent C (C: water with 0.1% formic acid) and solvent D (D: ACN containing 0.1% formic acid), separated by nano-LC, and analyzed online by electrospray tandem mass spectrometry. The experiment was conducted on an Easy-nLC 1000 system (Thermo Fisher Scientific, MA, USA), which was connected to an Orbitrap Fusion Lumos Tribrid mass spectrometer (Thermo Fisher Scientific, MA, USA), equipped with an online nano-electrospray ion source [141,142,143]. A 10 μL peptide sample was loaded onto a trap column (Thermo Scientific Acclaim PepMap C18, 100 μm × 2 cm) at a flow rate of 10 μL/min for 3 min, followed by analysis on an analytical column (Acclaim PepMap C18, 75 μm × 15 cm) with a linear gradient from 3% to 32% D over 120 min, re-equilibrated under initial conditions for 10 min, with a column flow rate maintained at 300 nL/min. The electrospray voltage at the mass spectrometer inlet was set to 2 kV. The Tribrid mass spectrometer was operated in data-dependent mode, automatically switching between MS and MS/MS acquisition. Full-scan MS spectra (*m*/*z* 350–1550) with a mass resolution of 120K were acquired, followed by continuous high-energy collision dissociation (HCD) MS/MS scans with a resolution of 30 K. The isolation window was set to 1.6 Da, the AGC target was set to 400,000, the fixed first mass in MS/MS was set to 110, and dynamic exclusion of repeated microscans was recorded with a 45 s window.

### 4.11. Proteomic Quantification and Bioinformatics Analysis

In this study, Mascot 2.3.02 software was employed to identify and quantitatively analyze the raw mass spectrometry data for peptides and proteins. A *t*-test analysis was conducted to screen for differentially expressed proteins (DEPs) between samples, with screening criteria set at a *p*-value < 0.05 and an inter-sample quantitative ratio (Fold Change, FC) ≥ 1.2. The differentially expressed proteins were then clustered and analyzed using principal component analysis through the R package (http://www.r-project.org/, accessed on 23 May 2024). Additionally, the biological functions of differential proteins were examined, and the identified proteins were compared against the GO, KOG, and KEGG databases using Blastx to obtain annotation information.

## 5. Conclusions

Transcriptome and proteome sequencing analysis was conducted on susceptible (S) strain and two resistant (R6, R14) strains of *L. pratensis* to identify key enzymes involved in detoxification, excavate genes and proteins associated with resistance, and screen transcription factor families related to detoxification metabolism. The co-up-regulated DEGs and DEPs were enriched in multiple detoxification metabolic signal pathways, including sphingolipid metabolism, Terpenoid backbone biosynthesis, ABC transporters, RNA transport, and Peroxisomes; the co-down-regulated DEGs and DEPs were mainly involved in Oxidative phosphorylation, Fatty acid biosynthesis, and Neuroactive ligand–receptor interaction, which regulate growth and development-related pathways, indicating that under the selection pressure of lambda-cyhalothrin, the activity of enzymes related to insect growth and development was inhibited. *L. pratensis* enhances its resistance to insecticides through complex physiological and biochemical changes, including transcription of detoxification genes, regulation of energy metabolism, and enhancement of antioxidant defense. Additionally, the gene annotated with Cytochrome *CYP6A13* was up-regulated at both transcriptional and translation levels, suggesting that the gene overexpression may be involved in the detoxification metabolism against lambda-cyhalothrin in *L. pratensis*. In this study, based on the combined analysis of transcriptome and proteome, we deeply explored the mechanism of detoxification metabolism against pyrethroid insecticides of *L. pratensis* and provided a theoretical basis for the development of pest management strategies.

## Figures and Tables

**Figure 1 ijms-26-01745-f001:**
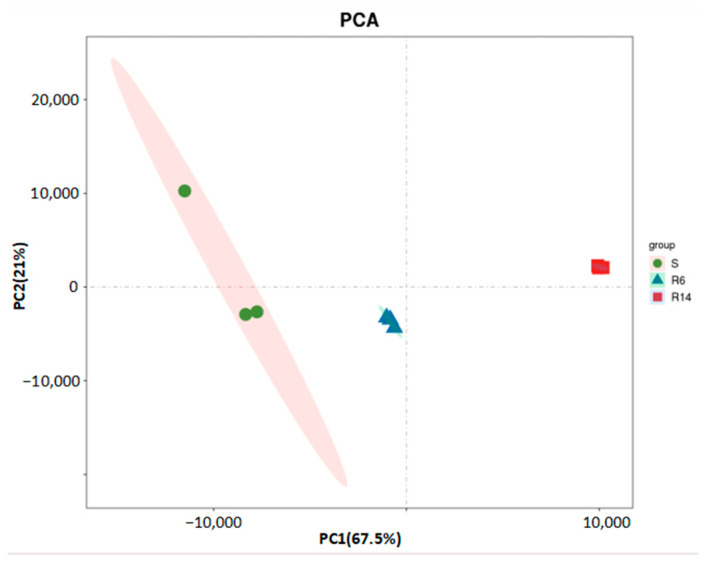
Principal component analysis (PCA) of transcriptome data.

**Figure 2 ijms-26-01745-f002:**
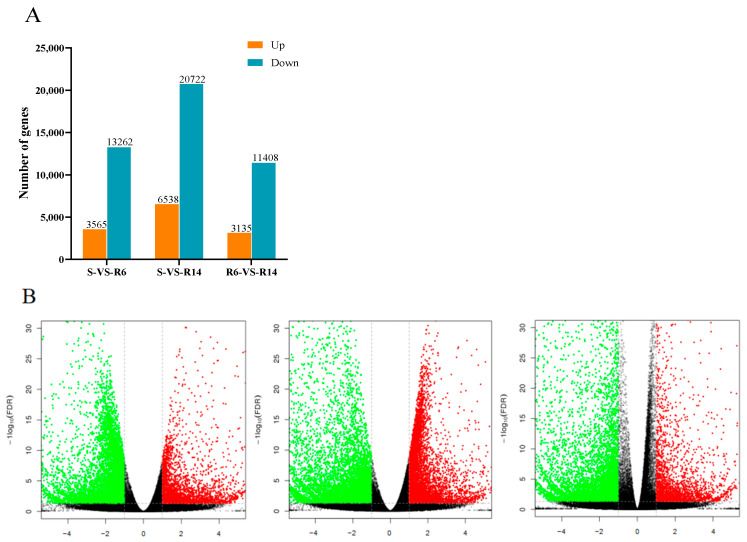
The number of significantly up-regulated and down-regulated genes between lambda-cyhalothrin-resistant and -susceptible strains of *L. pratensis*: (**A**) differential gene histogram; and (**B**) differential gene volcano map.

**Figure 3 ijms-26-01745-f003:**
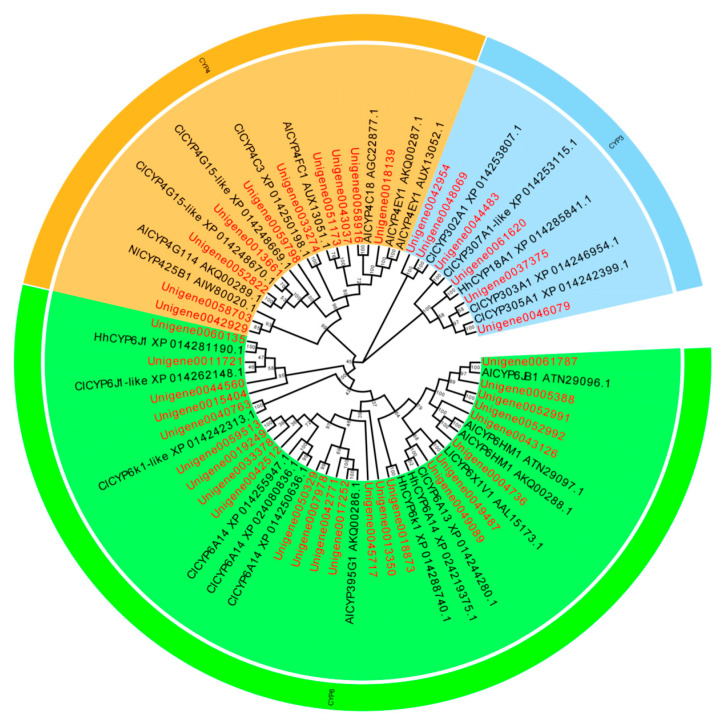
Phylogenetic evolutionary tree of the Cytochrome P450 gene of *L. pratensis* (red) with other insects (black). Note: The amino acid sequences of *Cimex lectularius* (Cl), *Apolygus lucorum* (Al), *Halyomorpha halys* (Hh), and *Nesidiocoris tenuis* (Nl) were analyzed using the MEGA-11 adjacency method. Green is a CYP6 family branch, orange is a branch of the CYP4 family, and blue is a branch of the CYP2 family. The two letters before the gene names are the species name. The topology was tested using bootstrap analyses (1000 replicates), and the number on the branch indicates the confidence coefficient.

**Figure 4 ijms-26-01745-f004:**
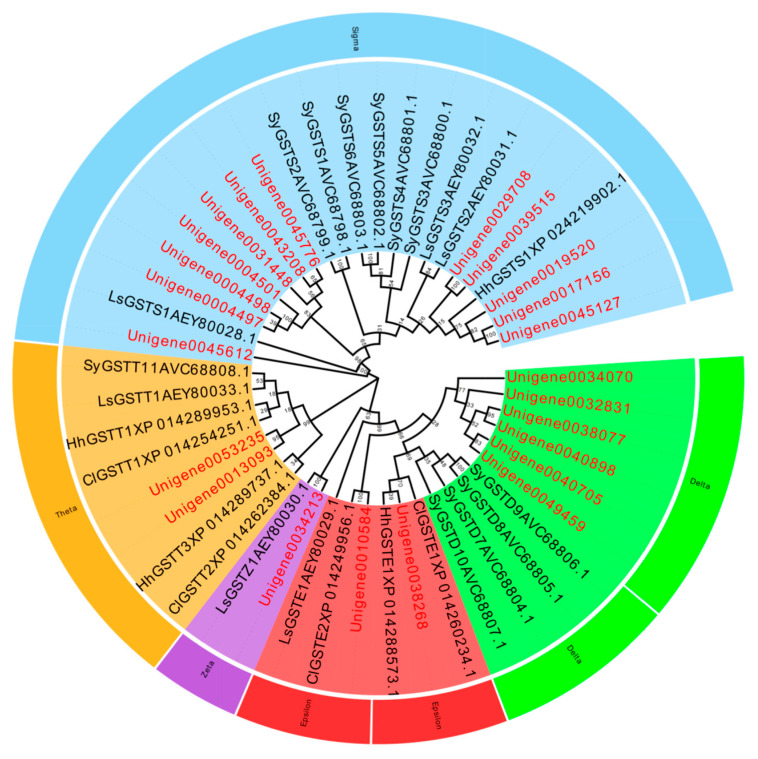
Phylogenetic evolutionary tree of Glutathione S-transferase genes of *L. pratensis* (red) with other insects (black). Note: The amino acid sequences of *Cimex lectularius* (Cl), *Halyomorpha halys* (Hh), *Laodelphax striatellus* (Ls), and *Subpsaltria yangi* (Sy) were analyzed using the MEGA-11 adjacent method. Green is a branch of the GST-Delta (D) family, red is a branch of the GST-Epsilon (E) family, purple is a branch of the GST-Zeta (Z) family, orange is a branch of the GST-Theta (T) family, and blue is a branch of the GST-Sigma (S) family. The two letters before the gene names are the species name. The topology was tested using bootstrap analyses (1000 replicates), and the number on the branch indicates the confidence coefficient.

**Figure 5 ijms-26-01745-f005:**
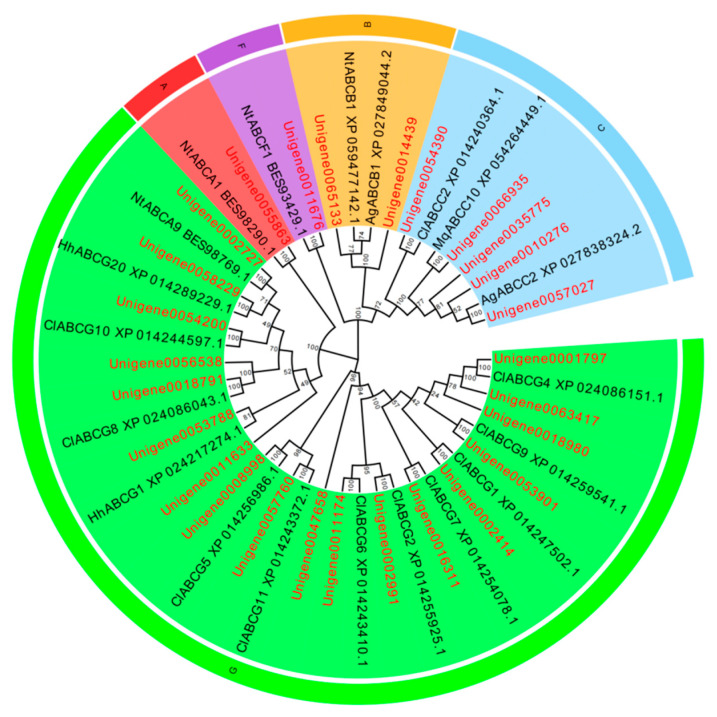
Phylogenetic evolutionary tree of ABC transporter genes from *L. pratensis* (red) with other insects (black). Note: The amino acid sequences of *Cimex lectularius* (Cl), *Halyomorpha halys* (Hh), *Macrosteles quadrilineatus* (Mq), and *Nesidiocoris tenuis* (Nt) were analyzed using the MEGA-11 adjacency method. Red is a branch of the ABC-A family, orange is a branch of the ABC-B family, blue is a branch of the ABC-C family, purple is a branch of the ABC-F family, and green is an ABC-G family branch. The two letters before the gene names are the species name. The topology was tested using bootstrap analyses (1000 replicates), and the number on the branch indicates the confidence coefficient.

**Figure 6 ijms-26-01745-f006:**
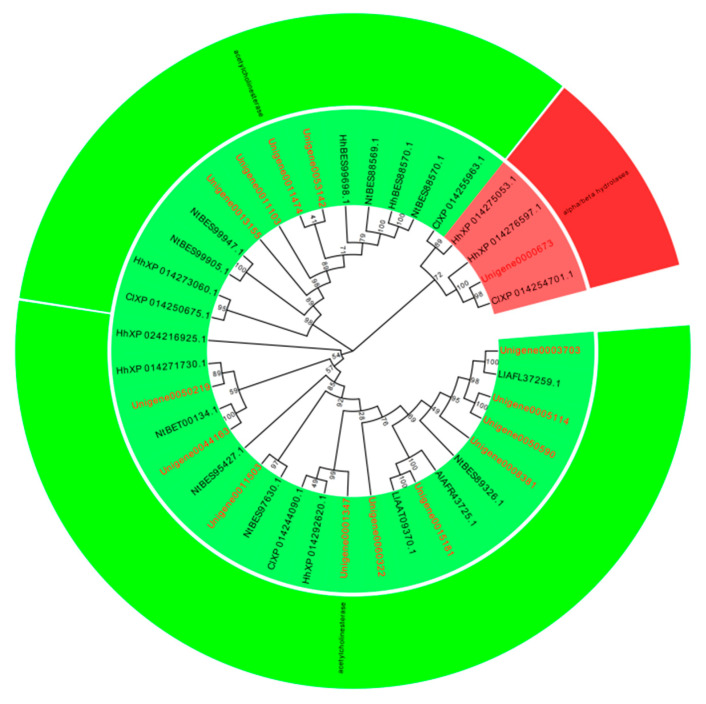
Phylogenetic evolutionary tree of Carboxylesterase genes of *L. pratensis* (red) with the other insects (black). Note: The amino acid sequences of *Cimex lectularius* (Cl), *Halyomorpha halys* (Hh), *Nesidiocoris tenuis* (Nt), *Apolygus lucorum* (Al), and *Lygus lineolaris* (Ll) were analyzed using the MEGA-11 adjacency method. Green is a branch of the Acetylcholinesterase family, and red is a branch of the alpha/beta hydrolases family. The two letters before the gene names are the species name. The topology was tested using bootstrap analyses (1000 replicates), and the number on the branch indicates the confidence coefficient.

**Figure 7 ijms-26-01745-f007:**
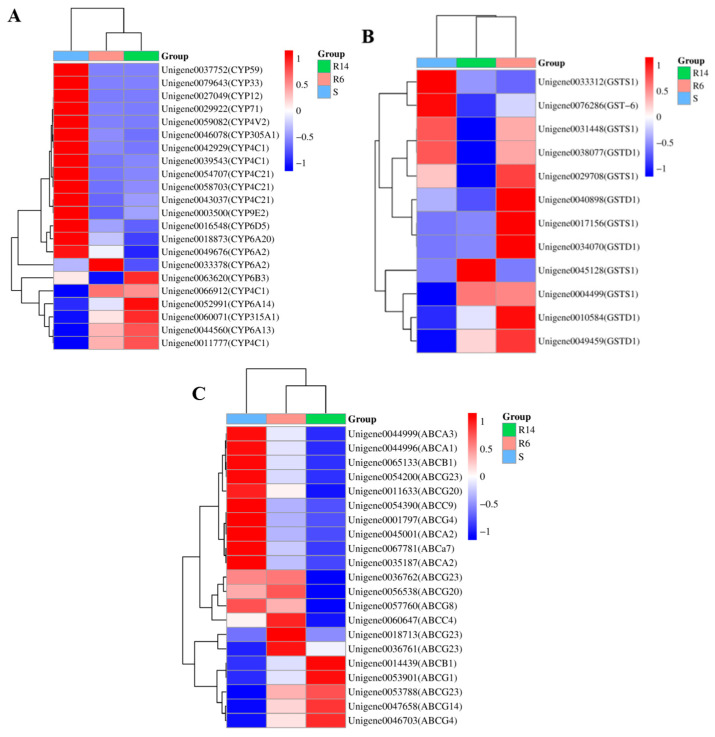
Analysis of differentially expressed genes associated with detoxification metabolism resistance: (**A**) Cytochrome P450 gene differential expression heat map; (**B**) Glutathione S-transferase gene differential expression heat map; and (**C**) ABC transporter gene differential expression heat map (red indicates up-regulation; blue indicates down-regulation).

**Figure 8 ijms-26-01745-f008:**
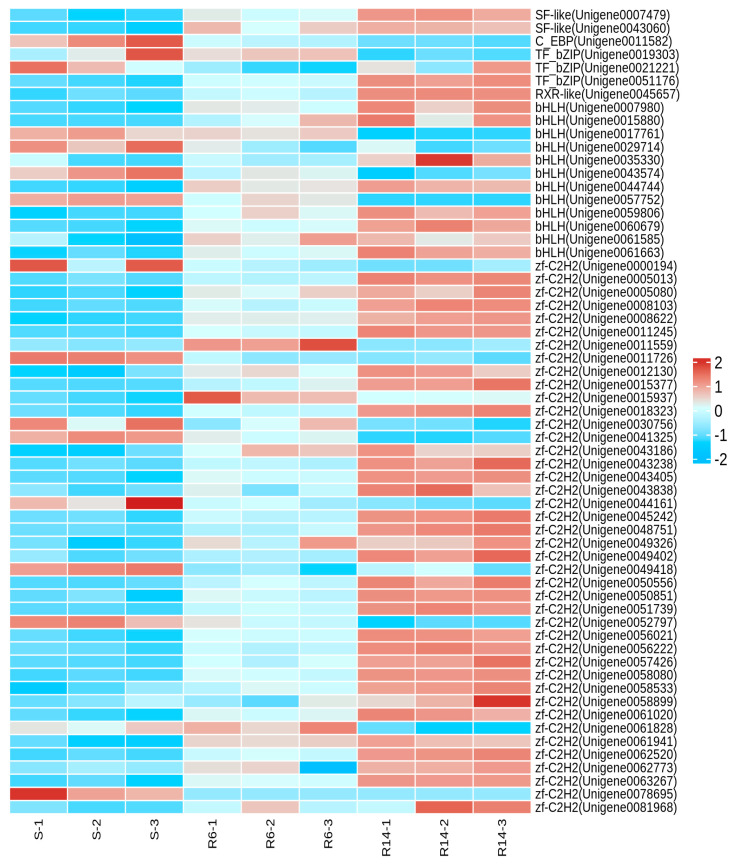
Heat map analysis of differentially expressed transcription factors.

**Figure 9 ijms-26-01745-f009:**
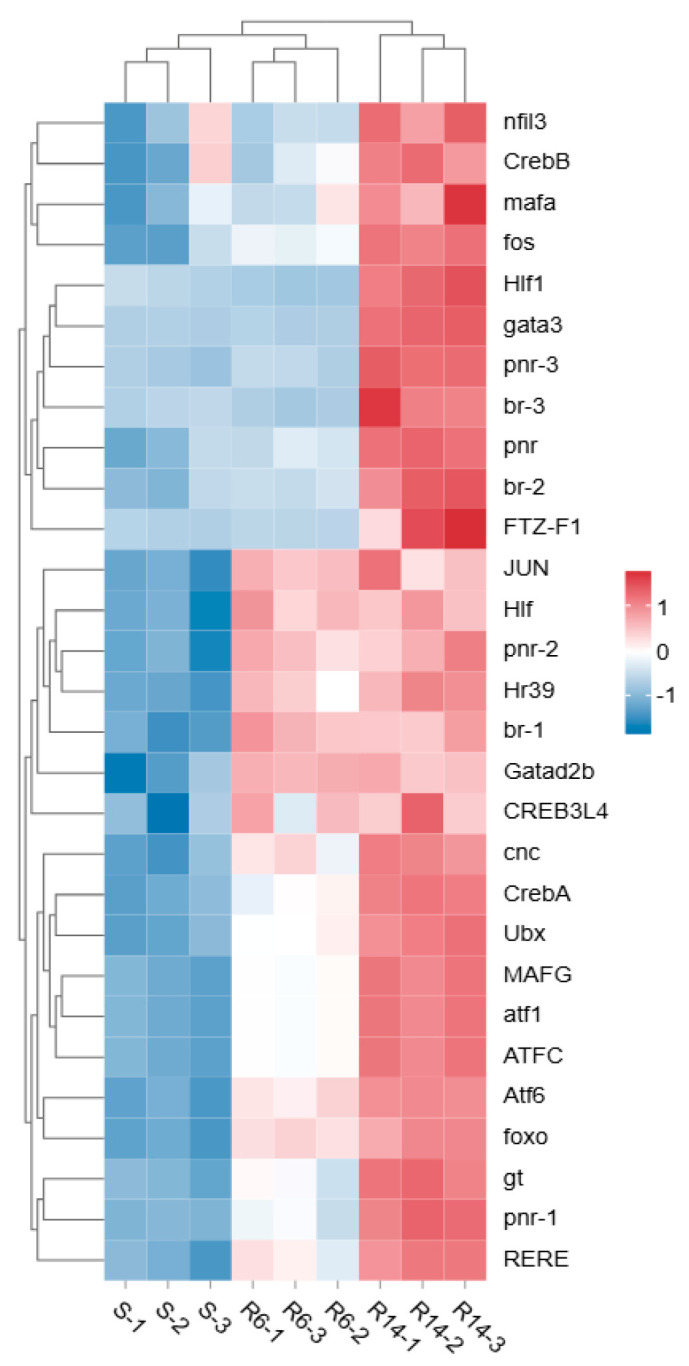
Heat map analysis of transcription factors associated with pyrethroid resistance.

**Figure 10 ijms-26-01745-f010:**
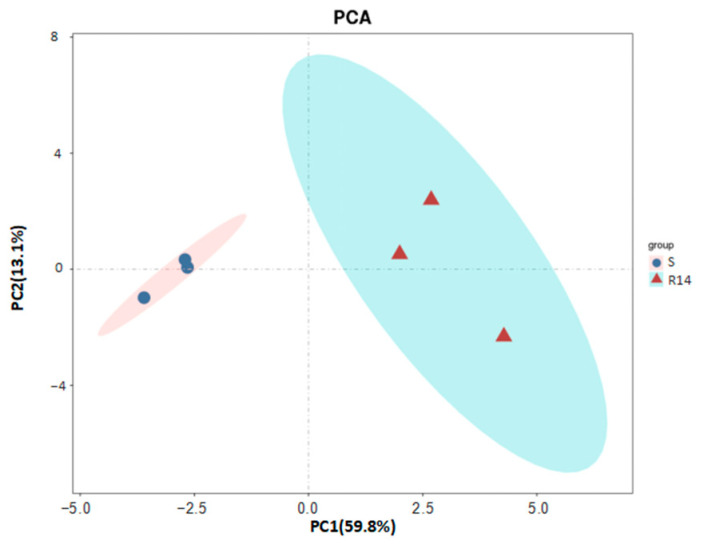
Principal component analysis (PCA) of proteome data.

**Figure 11 ijms-26-01745-f011:**
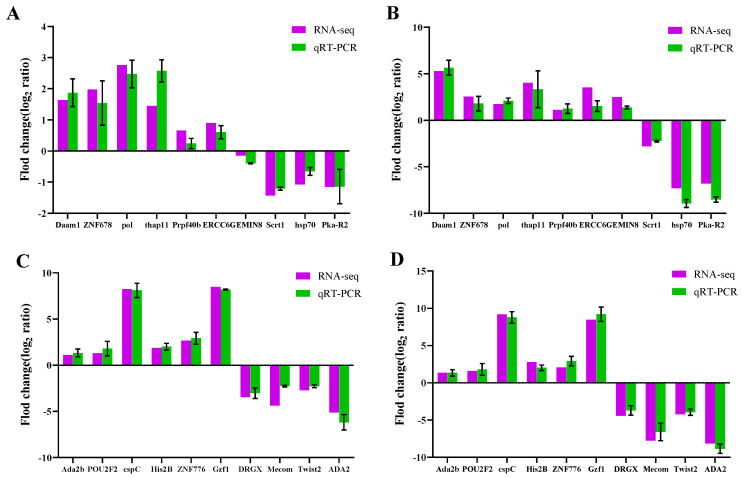
Validation of differentially expressed genes between RT-qPCR and RNA-Seq: (**A**) S-VS-R6; (**B**) S-VS-R14 (note: differentially expressed gene); (**C**) S-VS-R6; and (**D**) S-VS-R14 (note: differential expression of transcription factors).

**Figure 12 ijms-26-01745-f012:**
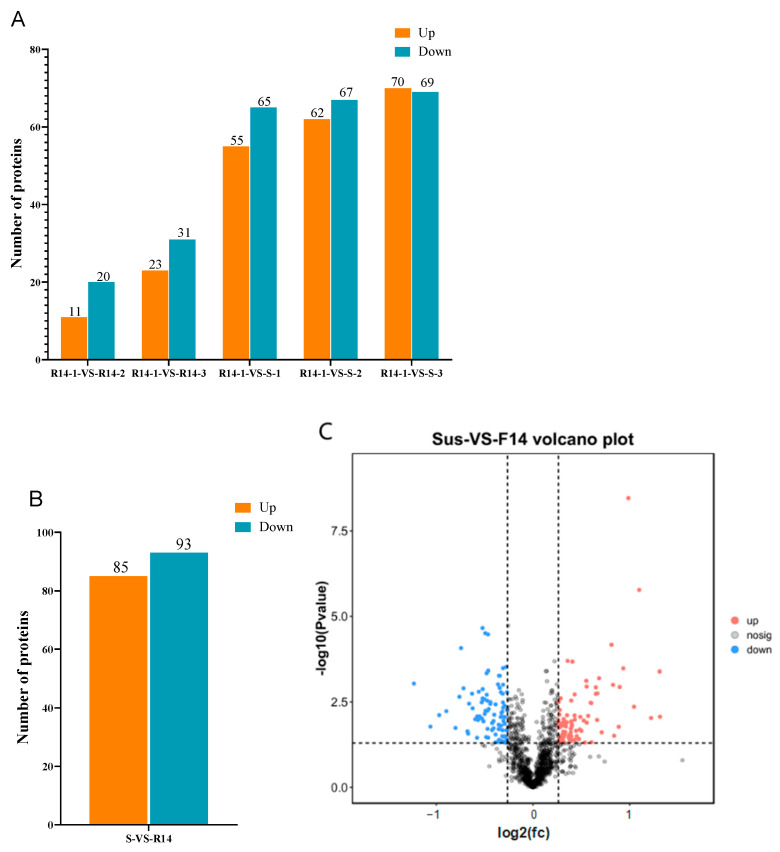
Statistical information of differently expressed proteins: (**A**,**B**) differential protein column diagrams; and (**C**) differential protein volcano diagram.

**Figure 13 ijms-26-01745-f013:**
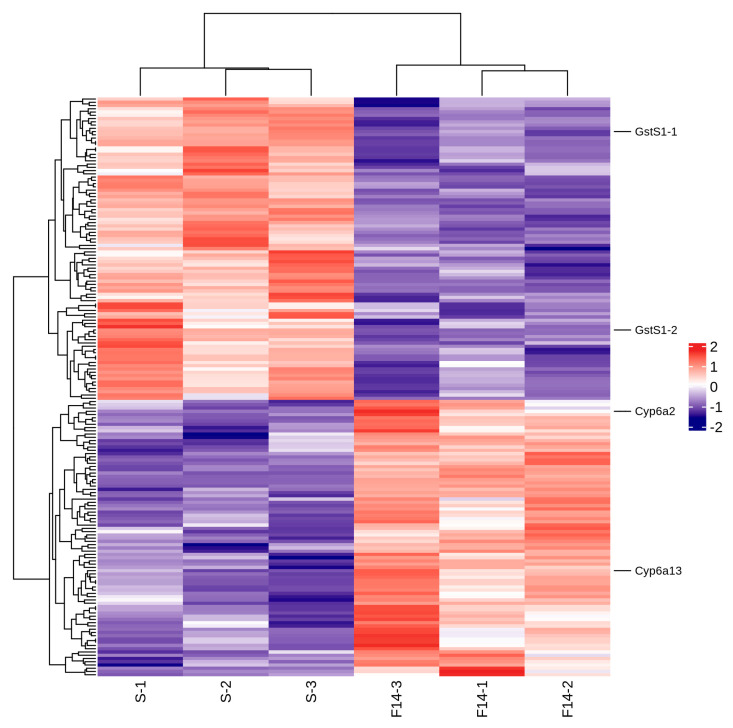
Proteomic differential protein heat map.

**Figure 14 ijms-26-01745-f014:**
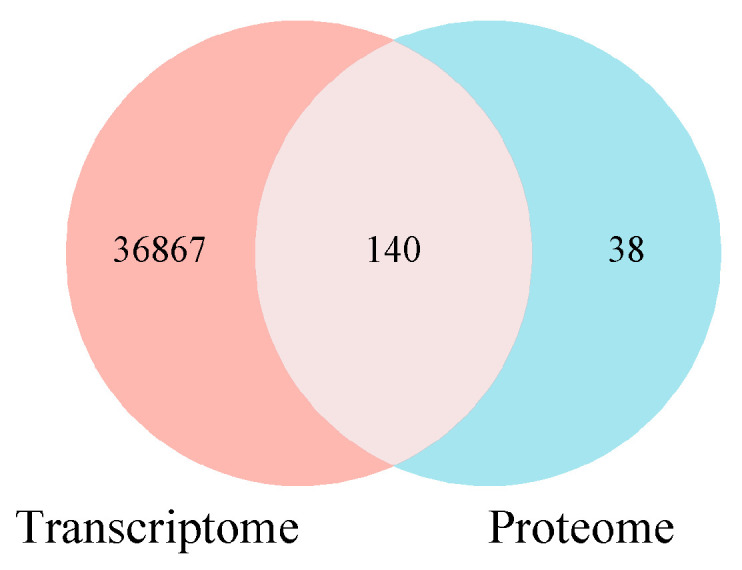
Venn diagram of differentially expressed genes and protein numbers. Note: Genes/proteins outside the threshold lines indicate significant differences, while those inside the threshold lines indicate non-significant differences.

**Figure 15 ijms-26-01745-f015:**
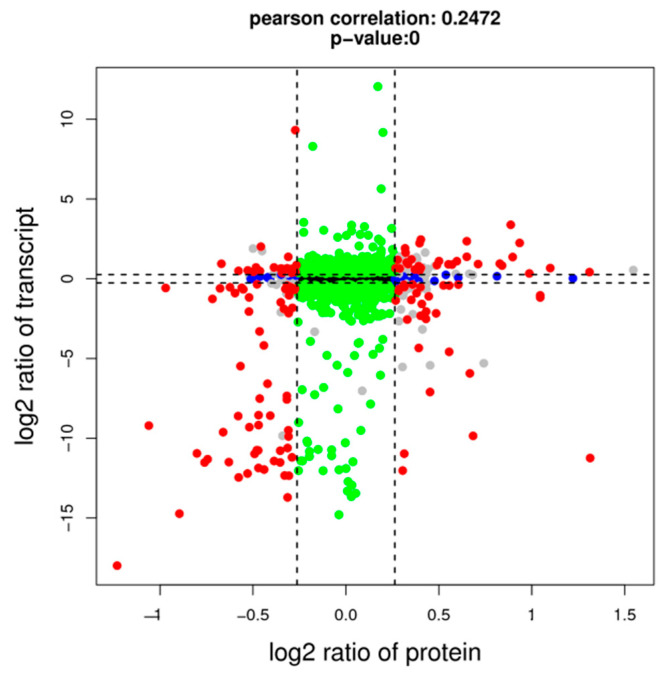
The nine-quadrant diagram analysis of transcriptome and proteome association. Note: Genes/proteins outside the threshold lines indicate significant differences, while those inside the threshold lines indicate non-significant differences. Red dots represent DEGs/DEPs, blue dots represent DEPs/NDEGs, green dots represent NDGEs/DEPs, black dots represent NDGEs/NDEPs, and gray dots represent DEGs/DEPs with *p*-value > 0.5.

**Figure 16 ijms-26-01745-f016:**
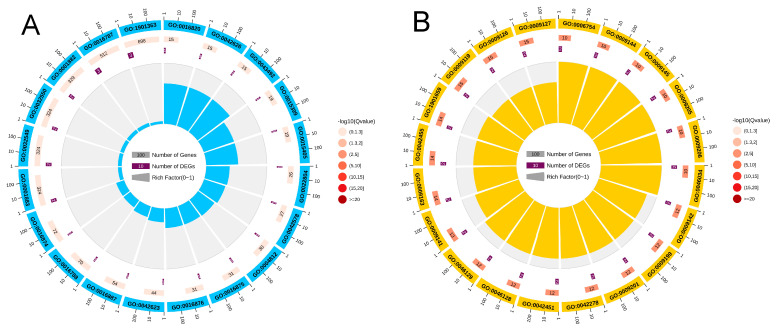
DEGs/DEPs GO enrichment classification: (**A**) co-up-regulated DEGs/DEPs GO enrichment cycle; (**B**) co-down-regulated DEGs/DEPs GO enrichment cycle; (**C**) co-up-regulated DEGs/DEPs GO enrichment column; and (**D**) co-down-regulated DEGs/DEPs GO enrichment column diagram.

**Figure 17 ijms-26-01745-f017:**
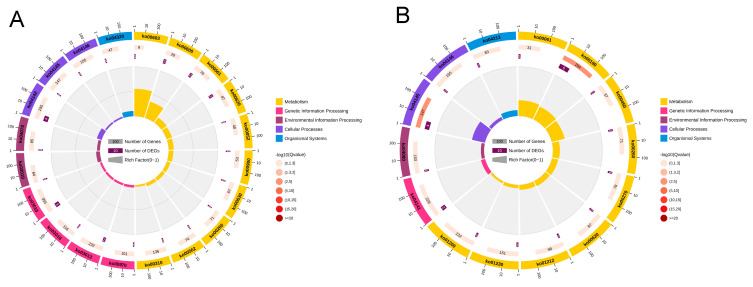
DEGs/DEPs KEGG enrichment classification: (**A**) co-up-regulated DEGs/DEPs KEGG enrichment cycle; (**B**) co-down-regulated DEGs/DEPs KEGG enrichment cycle; (**C**) co-up-regulated DEGs/DEPs KEGG enrichment column; and (**D**) co-down-regulated DEGs/DEPs KEGG enrichment column diagram.

**Table 1 ijms-26-01745-t001:** Analysis results of transcription factor family members of *Lygus pratensis*.

TF Family	Member Number	Percentage (%)	TF Family	Member Number	Percentage (%)	TF Family	Member Number	Percentage (%)	TF Family	Member Number	Percentage (%)
zf-C2H2	956	64.38%	RXR-like	9	0.60%	GCNF-like	4	0.27%	CSL	2	0.13%
Homeobox	70	4.67%	zf-GATA	7	0.47%	NGFIB-like	4	0.27%	AP-2	2	0.13%
THAP	50	3.34%	PAX	7	0.47%	NF-YC	4	0.27%	CP2	2	0.13%
MYB	43	2.87%	CSD	7	0.47%	SF-like	3	0.20%	CUT	2	0.13%
bHLH	42	2.80%	E2F	6	0.40%	C_EBP	3	0.20%	NCU-G1	1	0.06%
HTH	39	2.60%	Nrf1	6	0.40%	Tub	3	0.20%	DM	1	0.06%
HMG	31	2.10%	NF-YB	6	0.40%	zf-C2HC	3	0.20%	GCM	1	0.06%
TF_bZIP	22	1.47%	MH1	5	0.33%	AF-4	2	0.13%	CG-1	1	0.06%
Fork head	19	1.27%	zf-GAGA	5	0.33%	HMGI_HMGY	2	0.13%	CBF	1	0.06%
zf-LITAF-like	12	0.80%	T-box	5	0.33%	zf-NF-X1	2	0.13%	STAT	1	0.06%
zf-BED	12	0.80%	Pou	5	0.33%	Runt	2	0.13%	SAND	1	0.06%
ZBTB	12	0.80%	THR-like	5	0.33%	COE	2	0.13%	NF-YA	1	0.06%
TSC22	11	0.70%	P53	4	0.27%	SRF	2	0.13%	ESR-like	1	0.06%
ARID	10	0.67%	Miscellaneous	4	0.27%	RFX	2	0.13%	PC4	1	0.06%
ETS	10	0.67%	zf-MIZ	4	0.27%	HPD	2	0.13%	NDT80_PhoG	1	0.06%
MBD	9	0.60%	RHD	4	0.27%	TEA	2	0.13%	HSF	1	0.06%

**Table 2 ijms-26-01745-t002:** The statistics of differentially expressed transcription factors in different strains of *L. pratensis*.

Family	S-VS-R6		S-VS-R14	
	Up	Down	Up	Down
zf-C2H2	126	30	438	41
Homeobox	7	7	9	25
THAP	7	1	21	3
MYB	4	12	13	13
bHLH	6	1	11	8
HTH	2	5	9	5
HMG	3	7	10	7
TF_bZIP	0	3	2	5
other	29	19	85	39

**Table 3 ijms-26-01745-t003:** Analysis of transcription factors related to pyrethroid resistance of *L. pratensis*.

GeneID	TF Family	Name	Description
Unigene0009234	TF_bZIP	cnc	Fork head box protein O isoform X3 [*Cimex lectularius*]
Unigene0017754	TF_bZIP	CrebA	cyclic AMP response element-binding protein A [*Cimex lectularius*]
Unigene0019269	TF_bZIP	Atf6	cyclic AMP-dependent transcription factor ATF-6 alpha isoform X1 [*Halyomorpha halys*]
Unigene0019303	TF_bZIP	nfil3	nuclear factor interleukin-3-regulated protein [*Halyomorpha halys*]
Unigene0021221	TF_bZIP	Hlf	basic leucine zipper transcriptional factor ATF-like 3 [*Cimex lectularius*]
Unigene0036897	TF_bZIP	MAFG	transcription factor MafG-like [*Riptortus pedestris*]
Unigene0051176	TF_bZIP	atf1	cyclic AMP-dependent transcription factor ATF-2 isoform X2 [*Cimex lectularius*]
Unigene0052524	TF_bZIP	mafa	transcription factor MafB [*Cimex lectularius*]
Unigene0052926	TF_bZIP	ATFC	cyclic AMP-dependent transcription factor ATF-4 isoform X2 [*Cimex lectularius*]
Unigene0059129	TF_bZIP	gt	cell death specification protein 2-like [*Cimex lectularius*]
Unigene0060689	TF_bZIP	JUN	transcription factor AP-1 [*Halyomorpha halys*]
Unigene0062094	TF_bZIP	CrebB	cAMP-responsive element modulator isoform X3 [*Cimex lectularius*]
Unigene0062911	TF_bZIP	CREB3L4	cyclic AMP-responsive element-binding protein 3-like protein 1 isoform X1 [*Cimex lectularius*]
Unigene0068894	TF_bZIP	Hlf	thyrotroph embryonic factor-like isoform X4 [*Halyomorpha halys*]
Unigene0068964	TF_bZIP	fos	transcription factor kayak-like isoform X2 [*Cimex lectularius*]
Unigene0000842	zf-GATA	pnr	transcription factor GATA-4-like [*Cimex lectularius*]
Unigene0000842	zf-GATA	pnr	transcriptional repressor p66-beta [*Halyomorpha halys*]
Unigene0018296	zf-GATA	Gatad2b	GATA-binding factor C-like isoform X2 [*Cimex lectularius*]
Unigene0025381	zf-GATA	gata3	GATA-binding factor A-like isoform X2 [*Cimex lectularius*]
Unigene0031491	zf-GATA	pnr	GATA-binding factor A-like isoform X1 [*Cimex lectularius*]
Unigene0042433	zf-GATA	pnr	arginine-glutamic acid dipeptide repeats protein [*Cimex lectularius*]
Unigene0065661	zf-GATA	RERE	nitrogen regulatory protein GLN3-like isoform X2 [*Halyomorpha halys*]
Unigene0068266	zf-GATA	pnr	broad-complex core protein isoforms 1/2/3/4/5 isoform X5 [*Cimex lectularius*]
Unigene0041688	ZBTB	br	protein bric-a-brac 2-like [*Cimex lectularius*]
Unigene0046261	ZBTB	br	protein bric-a-brac 2 isoform X1 [*Cimex lectularius*]
Unigene0053826	ZBTB	br	nuclear hormone receptor FTZ-F1 isoform X3 [*Cimex lectularius*]
Unigene0007479	SF-like	FTZ-F1	nuclear hormone receptor FTZ-F1 beta [*Cimex lectularius*]
Unigene0043060	SF-like	Hr39	homeotic protein ultrabithorax-like isoform X1 [*Cimex lectularius*]
Unigene0037788	Homeobox	Ubx	nuclear factor erythroid 2-related factor 2 isoform X1 [*Cimex lectularius*]
Unigene0019512	Fork_head	foxo	transcription factor GATA-4-like [*Cimex lectularius*]

## Data Availability

The raw data supporting the conclusions of this article will be made available by the authors without undue reservation.

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
