# Peer review of "Integrative Analysis of Transcriptomics and Proteomics for Screening Genes and Regulatory Networks Associated with Lambda-Cyhalothrin Resistance in the Plant Bug Lygus pratensis Linnaeus (Hemiptera: Miridae)"

_ijms, 2025, doi:10.3390/ijms26041745_

Round 1
Reviewer 1 Report
Comments and Suggestions for Authors
1. L.77: The name of the taxon authority should not be mentioned and should not be in italics.
2. L.143-145: I expected that the authors confirmed the lack of insecticides on the collected Lygus pratensis insects. The confirmation should be through a GC analysis of some randomly selected insects.
3. L.145: What were the “remaining insects”?
4. L. 146: How can one be sure that in 6 or 14 generations a resistant strain should emerge?
5. L.156: What are the treatments?
6. L. sessions 2.2 and 2.3: The authors do not provide the company name and country for all commercial products they use. This omission exists in many commercial products apart from these two sessions. A reference should be given where the product is used.
7. L. Table S1: The Gene name of the β-tubulin gene is not given.
8. L. All tables in the supplementary material: There is no adequate description of the column names.
9. L.252: A reference is needed.
10. L.289: What R package was used?
11. L.304-307: Genes with lengths 200-300 were not more numerous since genes with lengths 300-400 and 400-500 had the same highest numbers.
12. L.324: A reference is needed.
13. L.321-327: Unclear and the text contradicts the findings in Fig.S3.
14. L.336: Confidence limits are not shown in Fig.2A.
15. L.391-392: The confidence numbers on the evolutionary diagram are not visible in Fig. 3 and 4.
16. Sessions 3.1.7 & 3.1.8: The sessions' titles refer to DEGs related to metabolic detoxification but the content clarifies the distinction. I think it is confusing for the reader.
17. L.43: What are the variables with ‘Sus’ and what are the individuals of the principal component? Ideally, a reduced set of the 2 or 3 records of the matrix to be analyzed would suffice in order to get an idea of what table is analyzed. Also, PCA is not a cluster analysis and the good separation should be followed by an indication of the variable that contributed most to this separation.
18. L.232: Is the term ‘susceptible’ mentioned with the ‘S’ abbreviation or ‘Sus’?
19. L.105108, page 5: Where are the red, green, blue, grey, or black in the Venn diagram?
20. L.357-358 (in Conclusion): Terms like ‘may be involved’ should be avoided in the MS. After all these analyses by employing several packages, routines, and functions the existence and role in the detoxification of insecticides in Lygus pratensis should be confirmed.
The entire MS should be rewritten with a detailed description of the employed procedures and a description of the packages where these functions reside. Moreover, the issues in the Discussion session should be based on the findings of this work. Terms like “should be” and “may be involved” or similar phrases should be avoided.
Author Response
L.77: The name of the taxon authority should not be mentioned and should not be in italics.
Response 1: With regard to the question that "the authoritative name of species identification should not be mentioned and should not be used in italics", we attached great importance to this format requirement and have carried out a comprehensive review and revision.
L.143-145: I expected that the authors confirmed the lack of insecticides on the collected Lygus pratensis insects. The confirmation should be through a GC analysis of some randomly selected insects.
Response 2: Thank you for the question about confirming that the collected insect samples have not been exposed to insecticides. Your suggestion is very pertinent. The insect samples we collected were from alfalfa fields at the experimental station in Helinger County, Hohhot, Inner Mongolia, where no chemical insecticides had ever been used prior to sample collection historically, the experimental station was a breeding base for alfalfa and never have sprayed pesticides. (See our previous research paper “Tan, Y.; Ma, Y.; Jia, B., et. al. Laboratory selection, cross-resistance, risk assessment to lambda-cyhalothrin resistance, and monitoring of insecticide resistance for plant bug Lygus pratensis (Hemiptera: Miridae) in farming-pastoral ecotones of northern China. J. Econ. Entomol. 2021, 114, 891–902.”) When selecting the sampling sites, we had fully considered environmental factors to make sure that the insects were not contaminated by insecticides. After collection, the insects were reared in the laboratory without the use of any insecticides. We used pesticide-free green bean pods as the food source and reared the insects in a strictly controlled environment to avoid any accidental chemical exposure.
Before the experiment, we conducted detailed observations and preprocessing of the insects to ensure that they exhibited normal behavior and physiology without any signs of insecticide exposure.
Although you suggested further confirmation through gas chromatography (GC) analysis to confirm that the insects had not been exposed to insecticides, we consider that our current sampling and processing methods are sufficiently rigorous to guarantee the reliability of the results. Moreover, considering that our research focus is to reveal the insecticide resistance mechanisms through RNA-seq and iTRAQ analyses, the initial purity of the samples has already been ensured through strict field collection and laboratory control.
L.145: What were the “remaining insects”?
Response 3: This expression may not be clear enough and could lead to misunderstandings. In our study, the term “remaining insects” actually refers to the insects that were selected for insecticide resistance. We revised it in the revised manuscript.
- 146: How can one be sure that in 6 or 14 generations a resistant strain should emerge?
Response 4: Thank you for your question. The meaning of the original text is ambiguous, we rewrote it as followed:
The resistant strain was originally collected from Helin County Experimental Station of the Institute of Grassland Research in April 2015, and maintained in the laboratory and selected by insecticide-film method with lambda-cyhalothrin for more than 20 generations, finally showing higher 90-fold resistance in laboratory tests. The resistant populations were continuously selected for 6 (R6, 9.33 ng a.i./adult, 7.5 fold) and 14 (R14, 55.03 ng a.i./adult, 42.5 fold) generations, the healthy virgin adults were collected as lambda-cyhalothrin resistant samples used for downstream experiment.
Our experimental design was based on the classic insecticide selection method. The definition of resistant strain was well-founded, as many research results: Afzal et al. (2018) reported that resistance rapidly increased in the early stages of continuous selection and then stabilized when selecting with deltamethrin on Phenacoccus solenopsis. Ma et al. (2021) observed a significant increasing in resistance at the 6th generation and stabilization at the 14th generation under continuous selection pressure. Many examples validated the definition: Resistant strain refers to the collection of biological organisms that are tolerant (resistant) to insecticides.
L.156: What are the treatments?
Response 5: Thank you for your question regarding “treatments.” The term “treatments” refers to the different experimental groups in our study, including the susceptible strain (S) and two resistant samples (R6 and R14). To clarify this, we removed ambiguous, redundant expression and revised “treatments” into “groups” in manuscript.
- sessions 2.2 and 2.3: The authors do not provide the company name and country for all commercial products they use. This omission exists in many commercial products apart from these two sessions. A reference should be given where the product is used.
Response 6: In response to your comment on the incomplete information of commercial products, we have made comprehensive revisions and supplements. The missing information mentioned in sections 1.2 and 2.2 has been fully supplemented and perfected.
- Table S1: The Gene name of the β-tubulingene is not given.
Response 7: With regard to your mention of "the specific name of the β-tubulin gene missing in Schedule S1", we confirm that the β-tubulin gene was identified and validated as the stably-expressed reference gene as the reference reported Jia et al (2019).
- All tables in the supplementary material: There is no adequate description of the column names.
Response 8: We have taken this suggestion very seriously and have provided detailed supplements and explanations for the column names in the tables to make sure that readers can better understand and utilize the data.
L.252: A reference is needed.
Response 9: Thank you for reminding us about the missing reference on line 252. We have now added the appropriate references to support the experimental methods and instrumentation used in our study.
L.289: What R package was used?
Response 10: In our research, we performed PCA analysis using the ggplot2 package to visualize the distribution and separation of transcriptomic data among different samples.
L.304-307: Genes with lengths 200-300 were not more numerous since genes with lengths 300-400 and 400-500 had the same highest numbers.
Response 11: Regarding the issue of gene length distribution that you raised, we have conducted a detailed review and analysis. According to the latest data, the number of unigenes with lengths of 200-300 bp is indeed the highest. Our finding is consistent with the description in the main text.
Explanation: This is because Trinity software attempts to combine short sequence fragments into longer Unigenes during assembly. If the final assembly results show the highest number of Unigenes in the 200-300 bp range, it indicates that the software has performed well in processing short sequences and can efficiently assemble them into meaningful gene fragments, rather than over-merging or losing (see the reference Jeffrey A. M., Wang Z., Next-generation transcriptome Assembly. Nature Reviews, Genetics. 2011, 12: 671-682.).
L.324: A reference is needed.
Response 12: We have added appropriate references to that matches the content here.
L.321-327: Unclear and the text contradicts the findings in Fig. S3.
Response 13: Thank you for pointing out that the content in lines 321-327 is unclear and contradictory to the findings in Figure S3. After careful examination and re-analysis, we have rewrote the content about Fig S3, as follows:
The results showed that 24,687 unigenes (29.8%) of L. pratensis were annotated in the Nr database. Among them, the number of homologous sequences matched with Cimex lectularius (53.1%, 7,233) is the highest, followed by Halyomorpha halys (20.4%, 2,783) and Lasius niger (8.1%, 1,103). The number of homologous sequences matched with other species, such as Anoplophora glabripennis, Priapulus caudatus, Diaphorina citri, Exaiptasia castaneum, Tribolium castaneum, and so on, are all below 500 (SI Appendix Figure S3). Figure 3 further showed the distribution of homologous sequences of these species, indicating that L. pratensis has a closer evolutionary relationship with Cimex lectularius and Halyomorpha halys.
L.336: Confidence limits are not shown in Fig.2A.
Response 14: In our study, the statistical method primarily relies on DESeq2. This method typically provides the expression differences of each gene and their significance (such as p-values and q-values adjusted by FDR), but it does not directly offer confidence intervals for the number of differentially expressed genes. The number of differentially expressed genes is a summary statistic based on multiple genes, rather than a direct measurement of a single gene. Therefore, the Y axis is specific number rather than statistical number.
L.391-392: The confidence numbers on the evolutionary diagram are not visible in Fig. 3 and 4.
Response 15: Thank you for pointing out the issue of the display of confidence numbers in Figure group of evolutionary diagrams. We have already labeled the confidence numbers on the branches of the phylogenetic trees in fact. Besides, we added explanation into the notes in Figure 3-6:
The topology was tested using bootstrap analyses (1000 replicates), and the number on the branch indicates confidence coefficient.
Sessions 3.1.7 & 3.1.8: The sessions' titles refer to DEGs related to metabolic detoxification but the content clarifies the distinction. I think it is confusing for the reader.
Response 16: Thank you very much for your attention to the ambiguous titles. We revised them “3.1.7 Identification and Phylogenetic Analysis of Detoxification Metabolism-Related Genes” and “3.1.8 Differential expression Analysis of Detoxification Metabolism-Related Genes” respectively.
L.43: What are the variables with ‘Sus’ and what are the individuals of the principal component? Ideally, a reduced set of the 2 or 3 records of the matrix to be analyzed would suffice in order to get an idea of what table is analyzed. Also, PCA is not a cluster analysis and the good separation should be followed by an indication of the variable that contributed most to this separation.
Response 17: The reviewers raised several questions about principal component analysis (PCA), including the meaning of the "Sus" variable, the definition of individuals in PCA analysis, and the interpretation of PCA results. 1. "Sus" stands for Susceptible, which is the term used to describe the control or untreated group in an experiment. In PCA analysis, the "Sus" variable refers to sample data from susceptible strains for comparison with other groups, such as resistant strains R6 and R14. These variables help us evaluate the differences between samples under different treatment conditions. We have changed "Sus" to "S", as in the original article. 2. Individual definition in PCA analysis, "individual" refers to the observed value of each sample. Specifically, these individuals were samples from different treatment groups, such as susceptible strains "Sus" and resistant strains "R6" and "R14". 3. You correctly point out that PCA is not cluster analysis, but a tool for reduction and data exploration. In our study, the main purpose of PCA was to reveal the differences among different treatment groups through principal component analysis. In order to explain the PCA results more clearly, we added explanation into the result section: Contribution of principal components: We will provide the variance explanation rate for each principal component (for example, PC1 explains 40% of the variance and PC2 explains 25% of the variance) and show how these principal components help us understand the structure of the data. We have made detailed revisions on lines 297-310 on page 8 and lines 456-470 on page 12.
The Principal Component Analysis (PCA) of six samples from two groups of L. pratensis was performed using the R package (https://ggplot2.tidyverse.org/) (R Foundation for Statistical Computing, Vienna, Austria), as shown in Figure 10: PC1 accounted for the largest variance in the proteomic data, representing 59.8% of the total variance. It primarily reflected the differences between the susceptible strain (S) and the 14th generation resistant strain (R14). On the PC1 axis, the S and R14 strain exhibited a complete separation trend, indicating significant differences in protein expression levels. PC2 accounted for the 2nd largest variance in the data, representing 13.1% of the total variance, and further distinguished the differences among the samples within the R14 strain. On the PC2 axis, the three replicates clustered closely together, indicating good reproducibility of the PCA analysis results. The PCA analysis results demonstrated significant differences between S and the R14 strain in terms of gene expression and protein expression levels. These differences are likely closely related to the development of lambda-cyhalothrin resistance. Additionally, the R14 strain exhibited good reproducibility at both the transcriptional and translational levels, further validating the reliability and consistency of data.
L.232: Is the term ‘susceptible’ mentioned with the ‘S’ abbreviation or ‘Sus’?
Response 18: Thank you very much for pointing out the abbreviation issue of the term “susceptible” on line 232 (L.232). The inconsistency you mentioned causes confusion for readers, and we checked all the writtings thoroughly. The “Sus” in the original text has been revised to “S” throughout to maintain consistency with the definition of “susceptible.”
L.105108, page 5: Where are the red, green, blue, grey, or black in the Venn diagram?
Response 19: Thank you for pointing out the issue with the legend placement in the Venn diagram on page 5 (L.105108). This was a layout error, and we sincerely apologize for it. We appreciate your careful observation. We have corrected the legend placement in the revised manuscript on page 45, lines 1325-1328.
L.357-358 (in Conclusion): Terms like ‘may be involved’ should be avoided in the MS. After all these analyses by employing several packages, routines, and functions the existence and role in the detoxification of insecticides in Lygus pratensis should be confirmed.
Response 20: Okay, according to your suggestions, we have revised the conclusion section on page 17, lines 706-708, to more explicitly convey the certainty and scientific nature of the research findings, while avoiding ambiguous wording.

Reviewer 2 Report
Comments and Suggestions for Authors
Chen et al., reported the integrative analysis of transcriptomics and proteomics to figure out the lambda-cyhalothrin resistance in Lygus pratensis. The results described multiple up- and down-regulated genes’ transcripts and gene’ translation in lambda-cyhalothrin resistant L. pratensis, focusing on detoxification metabolism pathway. This work shed the light on the way to elucidate the resistant mechanism of plant bugs against pesticide. But there are many confusions and brief descriptions.
1. How many generations of L. pratensis per year in alfalfa? Why do you select 6 and 14 generations for resistance analysis, but not more?
2. Pyrethroids target the voltage-gated sodium channel gene (VGSC), and mutations in this gene may result in knockdown resistance (kdr). When you analyse the RNA-seq data, do you also find the mutation in your target insect?
3. How many larvae in a single sample? Is there any effect of single larva or mixed samples on the omics analysis?
4. Can detail the tools and parameters that you used for raw filter?
5. Have you get the full length of resistance gene transcript that you proposed in ms?
6. Why do you pick up these genes, Cytochrome P450, GST, and ABC proteins? They are obvious different in resistance test.
7. In you Qpcr part you mentioned that you select ten genes for quantification, have you run the standard curve for these ten genes? As you know, the 2-ΔΔCt method require that the efficiency of Qpcr should be 95-105%.
8. In line 185A minor error: “A T-test analysis was” should be “A t-test analysis was”
9. In line 335,“The number of dif-335 ferential transcripts in R14 is significantly higher than in R6.” Cannot find any statistic there when you said significant difference.
10. Line 434, please italicize the Latin names “L. pratensis”. And other part across the whole ms.
11. What is the meaning of F14 in figure 11?
12. Figure 14, where is the dot in Figure 14.“Venn diagram of differentially-expression genes and protein numbers. Note: Genes/Proteins outside the threshold lines indicated significant differences, while those inside the threshold lines indicated non-significant differences. Red dots represent DEGs/DEPs, blue dots represent DEPs/NDEGs, green dots represent NDGEs/DEPs, black dots represent NDGEs/NDEPs, and gray dots represent DEGs/DEPs with P-value > 0.5”
The combined analysis of Transcriptomics and proteomics showed the co-up-regulated DEGs and DEPs and the co-down-regulated DEGs and DEPs were more than 300 genes. Although these were described data, they offer the basic information of plant bug against pyrethroid. I would like to recommend it publish in IJMS.
Author Response
ief descriptions.
- How many generations ofL. pratensis per year in alfalfa? Why do you select 6 and 14 generations for resistance analysis, but not more?
Response 1: Thank you for your question. "Why were the 6th and 14th generations chosen?" is based on the observations of resistance development and stabilization in our study. We chose the 6th and 14th generations for detailed analysis because the LD50 values have turning points after the two generations during the resistance selection process. Our experimental design was informed by previous research methods. For example, Afzal et al. (2018) found that resistance rapidly increased in the early stages of continuous selection and then stabilized when screening Phenacoccus solenopsis with deltamethrin. This is consistent with our observations in Lygus pratensis, where Ma (2021) observed a significant increase in resistance at the 6th generation and stabilization at the 14th generation under continuous selection pressure. We have made detailed modifications to the relevant content on page 4, line 126 to 137 of the manuscript, specifically describing the trends in resistance across different generations and citing relevant studies to support experimental design.
- Pyrethroids target the voltage-gated sodium channel gene (VGSC), and mutations in this gene may result in knockdown resistance (kdr). When you analyse the RNA-seq data, do you also find the mutation in your target insect?
Response 2: In our transcriptome analysis, we did not detect any transcripts directly related to the voltage-gated sodium channel gene (VGSC). This has to do with the strain. However, we detected the presence of kdr mutations in VGSC in other field populations, but not the susceptible and resistance strains, see Zhang et al. (2024.) “Zhang, L.Q.; Ni, R.Y.; Chen, J.; Yang, J.L.; Dong, Y.W.; Yuchi, Z.G.; Tan, Y. Molecular detection of kdr and superkdr mutation sites and analysis of the binding modes of pyrethroid insecticides with voltage-gated sodium channels in the plant bug Lygus pratensis (Hemiptera: Miridae). J. Agric. Food Chem. 2024, 10, 1021.”
- How many larvae in a single sample? Is there any effect of single larva or mixed samples on the omics analysis?
Response 3: In our research, each sample contains 500 larvae, all from the same treatment group, to make sure the representativeness and reproducibility of the samples. Using multiple larvae can reduce the impact of individual differences on the experimental results, thereby more accurately reflecting the differences among different groups. Additionally, using mixed samples can improve the sensitivity and accuracy of detection while reducing experimental costs.
- Can detail the tools and parameters that you used for raw filter?
Response 4: Thank you very much for your attention to our research and your valuable comments. We used FastQC for quality assessment and set the parameters in Trimmomatic with a quality threshold of Q20 and Q30, and a minimum read length of 200 bp. We have added the writing contents on page 4, lines 157-161. As followed:
To filter the raw data obtained from sequencing, we used FastQC for quality assessment and Trimmomatic for data filtering. The filtering process ensured that the resulting clean reads met the quality criteria of Q20 and Q30, with the minimum read length set at 200 bp. This stringent filtering produced high-quality clean reads to ensure the accuracy and reliability of the sequencing data.
- Have you get the full length of resistance gene transcript that you proposed in ms?
Response 5: Regarding your question about the full-length transcript of the resistance gene mentioned in the manuscript, we have already cited a previous study from our laboratory on page 12, line 508. In that study, the full-length cDNA sequence of the CYP6A13 gene was successfully cloned using RT-PCR and RACE technique (Ma, 2021). “Ma, Y.; Zhang, W.B.; Zhang, H.L.; Ma, H.Y.; Han, H.B.; Pang, B.P.; Tan, Y. Cloning of CYP6A13 and sodium channel gene LPVSSC and their involvement in analysis on the resistance of lambda-cyhalothrin in Lygus pratensis (Hemiptera: Miridae). Environ. Entomol. 2022, 44, 1252-1263.”
- Why do you pick up these genes,Cytochrome P450, GST, and ABC proteins? They are obvious different in resistance test.
Response 6: In our study, we selected Cytochrome P450, Glutathione S-transferase (GST), and ABC transporters as our focus family genes strongly associated with resistance based on substantial literatures that demonstrated close involvements between the significantly different expression of P450 genes, GST genes, and ABC transporter genes and insect resistance to pyrethroid insecticides (In our manuscript, many references were cited). The differences observed in the resistance research papers may be related to their distinct roles in detoxification mechanisms: Cytochrome P450 can enhance the insect’s metabolic detoxification capability by oxidatively metabolizing insecticides. Glutathione S-transferase (GST) can increase the water solubility of insecticides, facilitating their excretion and thereby enhancing detoxification. ABC transporters can reduce the accumulation of insecticides within cells, thereby lowering their toxicity. Although these proteins have different functions and action mechanisms, but many of which make significant contributions to insecticide resistance. Therefore, we pick up these genes.
- In your q-PCR part you mentioned that you select ten genes for quantification, have you run the standard curve for these ten genes? As you know,the 2-ΔΔCt method require that the efficiency of Qpcr should be 95-105%.
Response 7: Thank you very much for raising the issue regarding the standard curve in the qPCR experiment. To ensure the accuracy and reliability of the qPCR results, we ran standard curves for each of 10 selected genes and calculated the amplification efficiency in our experiment. We have added the missing information in a table on page 10, lines 401-410 in the main text, and on page 6, line 79 in the supplementary table.
- In line 185A minor error: “A T-test analysis was” should be “At-test analysis was”.
Response 8: We have already corrected this error in the revised manuscript.
- In line 335,“The number of dif-335 ferential transcripts in R14 is significantly higher than in R6.” Cannot find any statistic there when you said significant difference.
Response 9: You pointed out that in line 335, the statement “the difference between R14 and R6 is significant” lacks statistical evidence to support the claim of “significance.” Definitely, we removed the adverb.
- Line 434, please italicize the Latin names “L. pratensis”. And other part across the whole ms.
Response 10: We have conducted a thorough review and made the necessary corrections as per your advice.
- What is the meaning of F14 in figure 11.
Response 11: It should indeed be “R14,” consistent with the terminology used in the main text.
- Figure 14, where is the dot inFigure 14.“Venn diagram of differentially-expression genes and protein numbers.Note: Genes/Proteins outside the threshold lines indicated significant differences, while those inside the threshold lines indicated non-significant differences. Red dots represent DEGs/DEPs, blue dots represent DEPs/NDEGs, green dots represent NDGEs/DEPs, black dots represent NDGEs/NDEPs, and gray dots represent DEGs/DEPs with P-value > 0.5”.
Response 12: The last part of the figure legend for Fig. 14 has been moved to Fig. 15 to correct the association.

Round 2
Reviewer 1 Report
Comments and Suggestions for Authors
I thank the authors for the first reply. I have no further suggestions for the MS and I recommend the publication of the MS as is.